# THE SMART BUILDINGS CONTROL SUITE: A DIVERSE OPEN SOURCE BENCHMARK TO EVALUATE AND SCALE HVAC CONTROL POLICIES FOR SUSTAINABILITY

## ABSTRACT

Commercial buildings account for 17% of U.S. carbon emissions, with roughly half of that from Heating, Ventilation, and Air Conditioning (HVAC). HVAC devices form a complex thermodynamic system, and while model predictive control and reinforcement learning have been used to optimize control policies, scaling to thousands of buildings remains a significant unsolved challenge. Most current approaches are over-optimized for specific buildings and rely on proprietary data or hard-to-configure simulators. We present the Smart Buildings Control Suite, the first open source interactive HVAC control benchmark with a focus on solutions that generalize across building. It has 3 components: real-world data from 11 buildings over 6 years, a lightweight data-driven simulator for each building, and a modular Physically Informed Neural Network (PINN) building model as a simulator alternative. The buildings span multiple climates, management systems, and sizes, and both the simulator and PINN easily transfer to new buildings, ensuring solutions using this benchmark are robust to these factors and only reliant on fully scalable building models. This represents a major step towards scaling HVAC optimization from the lab to buildings everywhere. To facilitate use, our benchmark is compatible with the Gym standard, and our data is part of TensorFlow Datasets.

## 1 INTRODUCTION

Energy optimization and management in commercial buildings is a very important problem, whose importance is only growing with time. Buildings account for 37% of all US carbon emissions, with commercial buildings alone taking up a staggering 17% in 2023 (EIA). Reducing those emissions by even a small percentage can have a significant effect, especially in more extreme climates. We believe this problem is one of the most important avenues for climate sustainability research, where even a small improvement over baseline policies can drastically reduce our carbon footprint.

In particular, HVAC systems account for 40-60% of energy use in buildings (Pérez-Lombard et al., 2008) and roughly 15% of the world's total energy consumption (Asim et al., 2022). Most office buildings are equipped with advanced HVAC devices, like Variable Air Volumes (VAVs), Hot Water Systems, Air Conditioners (ACs) and Air Handlers (AHUs) that are configured and tuned by the engineers, manufacturers, installers, and operators to run efficiently with the device's local control loops (McQuiston et al., 2023). However, integrating multiple HVAC devices from diverse vendors into a building "system" requires technicians to program fixed operating conditions for these units, which may not always be optimal, and thus an ML model can be trained to continuously tune a small number of setpoints to achieve greater energy efficiency and reduced carbon emission.

Optimizing HVAC control has been an active research area for decades, and yet while AI has begun to transform many industries, to date almost all HVAC systems remain the same as they were 30 years ago: despite all the literature on the topic, no single solution has been widely adopted in the real world. One of the most significant factors limiting progress is the absence of a reliable public benchmark to evaluate solutions. Current efforts often rely on proprietary data and expensive closed-source simulations. This restricts participation to those with exclusive access and makes it

challenging to verify and compare results. Another major challenge is generalization across different buildings. Most solutions are tailored to specific buildings and fail to generalize for two reasons:

1. Complex building models: Many approaches rely on high-fidelity simulators or models that are difficult and time-consuming to configure for arbitrary buildings.
2. System-specific dependencies: Solutions are often tied to a particular BMS, HVAC system design, or data ontology, making them brittle and unable to adapt to the variability of other systems.

A large scale and diverse public benchmark would facilitate collaborations between institutions, standardize research efforts, allow for wider participation, and sharpen the focus on generalizable solutions. Much of the progress in AI has been driven by easily accessible public benchmarks, from the ImageNet Challenge in Vision (Russakovsky et al., 2015), to the Atari57 suite in Reinforcement learning (RL) (Badia et al., 2020), and the GLUE Benchmark in NLP (Wang et al., 2018). A similar benchmark in HVAC control may help accelerate progress and finally lead to real-world adoption.

We present The Smart Buildings Control Suite, a diverse, high quality, fully accessible, building control benchmark; the first of its kind. It consists of the following components:

1. Real-world historical HVAC data, collected from 11 buildings spanning 3 building management systems from across North America, over a 6-year period.
2. A highly customizable and scalable HVAC and building simulator that can be calibrated with real data, with configurations corresponding to each of the above buildings, as well as a pipeline for easily onboarding and calibrating new buildings.
3. A modularized neural network architecture incorporating physical priors for building energy modeling, as a fully data-driven, physically informed neural network (PINN) simulator alternative, as well as models trained to emulate each of the above buildings, and instructions for training a model on a new building.
4. A focus on ease of use. This includes full compatibility with the OpenAI Gym standard(Brockman et al., 2016) so that a model can be trained either on offline real data or interactively from the simulator or PINN backend. Our data is also available on the popular TensorFlow Datasets platform (TFDS), and our code is open source.

This benchmark advances us towards a future where machine learning research and building operations are working in tandem to decarbonize our buildings, a crucial step for humanity to address the climate crisis.

## 2 RELATED WORK

Considerable attention has been paid to HVAC control (Fong et al., 2006) in recent years (Kim et al., 2022), and while alternative approaches exist, such as Model Predictive Control (MPC) (Taheri et al., 2022), a growing portion of the literature has considered how RL and MPC can be leveraged (Yu et al., 2021; Mason & Grijalva, 2019; Yu et al., 2020; Gao & Wang, 2023; Wang et al., 2023; Vázquez-Canteli & Nagy, 2019; Zhang et al., 2019b; Fang et al., 2022; Zhang et al., 2019b; Goldfeder & Sipple, 2024). As mentioned above, a central requirement is an offline environment that trains the RL agent. Several methods have been proposed, largely falling under three broad categories. Our benchmark correspondingly has three parts, representing the state of the art of each.

**Offline RL on Real Data** The first approach is to train the agent directly from the historical real-world data, without ever producing an interactive environment (Chen et al., 2020; 2023a; Blad et al., 2022). While the real-world data is obviously of high accuracy and quality, this presents a major challenge, since the agent cannot take actions in the real world and interact with any form of an environment. This inability to explore severely limits its ability to improve over the baseline policy producing the real-world data (Levine et al., 2020). Furthermore, prior to our work, there are few public datasets available. Our dataset, with its diverse buildings and long duration, should allow for rapid development of offline RL agents in a way not previously possible.

**Data-driven Emulators** Some work attempts to learn dynamics as a multivariate regression model from real-world data (Zou et al., 2020; Zhang et al., 2019a), often using recurrent neural network architecture (Velswamy et al., 2017; Sendra-Arranz & Gutiérrez, 2020; Zhuang et al., 2023). The difficulty here is that data-driven models often do not generalize well outside the training distribution, especially since they are not physics-based. To overcome this limitation, PINNs incorporate

physical priors to enforce reasonable behavior in out-of-distribution scenarios Djeumou et al. (2022); Wang & Dong (2023); Chen et al. (2023b); Gokhale et al. (2022); Jiang & Dong (2024), and our PINN model builds off the state of the art in this area.

**Physics-based Simulation** HVAC system simulation has long been studied (Trčka & Hensen, 2010; Riederer, 2005; Park et al., 1985; Trčka et al., 2009; Husaunndee et al., 1997; Trcka et al., 2007; Blonsky et al., 2021). EnergyPlus (Crawley et al., 2001), a high-fidelity simulator developed by the Department of Energy, is commonly used (Wei et al., 2017; Azuatalam et al., 2020; Zhao et al., 2015; Wani et al., 2019; Basarkar, 2011), but suffers from scalability and configuration challenges. To overcome the limitations of these methods, some work uses a hybrid approach (Zhao et al., 2021; Balali et al., 2023; Goldfeder & Sipple, 2023; Zhang et al., 2023; Klanatsky et al., 2023; Drgoňa et al., 2021), as does our simulator. What is unique about our approach is the use of a physics-based simulator that achieves an ideal balance between speed of configuration, and fidelity to the real world. Our simulator is lightweight enough to be configured rapidly to an arbitrary building, easy to calibrate with data, and accurate enough to train an effective controller.

**Prior Datasets** While many building datasets exist (Ye et al., 2019), most have either a different focus (Sachs et al., 2012; Urban et al., 2015; Kriechbaumer & Jacobsen, 2018; Granderson et al., 2023), do not contain sufficient HVAC information (Miller et al., 2020; Mathew et al., 2015; Rashid et al., 2019; Jazizadeh et al., 2018; Sartori et al., 2023), focus on residential buildings (Murray et al., 2017; Barker et al., 2012; Meinrenken et al., 2020) or non-standard buildings (Pettit et al., 2014; Biswas & Chandan., 2022), or are simulated (Field et al., 2010; Bakker et al., 2022). The few datasets directly relevant (Luo et al., 2022; Heer et al., 2024) are non-interactive and come from a single building management system. We present the first HVAC control benchmark that has high quality real-world data from diverse sources, along with computationally cheap data-driven simulation and PINN building models, allowing for both real-world grounding and interactive control experiments.

# 3 OPTIMIZING ENERGY AND EMISSION IN COMMERCIAL BUILDINGS

## 3.1 PROBLEM FORMULATION

We frame energy optimization in buildings as a Markov Decision Process (MDP)(Garcia & Rachelson, 2013). We define the state of the building $S_t$ at time $t$ as a fixed length vector of measurements from sensors on the building's devices, such as a specific VAV's zone air temperature, gas meter's flow rate, etc. The action on the building $A_t$ is a fixed-length vector of device setpoints selected by the controller at time $t$, such as the boiler supply water temperature setpoint, etc. The controller observes the state $S_t$ from the environment at time $t$, then chooses action $A_t$. The environment responds by transitioning to the next state $S_{t+1}$ and returns a reward after the action, $R_{t+1}$.

Thus the MDP is formally described by the tuple $(S, A, p, R)$ where the state space is continuous (e.g., temperatures, flow rates, etc.) and the action space is continuous (e.g., setpoint temperatures) and the transition probability $p : S \times S \times A \rightarrow [0, 1]$ represents the probability density of the next state $S_{t+1}$ from taking action $A_t$ on the current state $S_t$. The reward function $R : S \times A \rightarrow [R_{min}, R_{max}]$ emits a scalar at each time $t$. The controller is acting under policy $\pi_\theta(A_t|S_t)$ parameterized by $\theta$ that represents the probability of taking action $A_t$ from state $S_t$. The MDP formalism is broad and allows for many optimization strategies, such as RL, rule based controls, and MPC. For an overview of approaches to learning an optimal policy, see Appendix A.

## 3.2 REWARD FUNCTION

The MDP formalism generally requires a single scalar reward signal, $R_t(S_t, A_t)$ that indicates the quality of taking action $A_t$ in state $S_t$. Since this is a multi-objective optimization problem, we define a custom feedback signal, $R_{3C}$, as a weighted sum of negative cost functions for carbon emission, energy cost, and comfort levels within the building, which we call the 3C Reward. It is governed by the following equation:

$$R_{3C} = u \times C_1 + v \times C_2 + w \times C_3$$

where $C_1$ represents normalized comfort conditions, $C_2$ normalized energy cost and $C_3$ normalized carbon emission. Constants $u$, $v$, $w$ represent operator preferences, allowing them to weigh the relative importance of cost, comfort, and carbon consumption. $R_{3C} = 0$ represents the best-case scenario: no energy is consumed, no carbon is emitted, and all occupied zones are in setpoint bounds. For details and equations governing how we measure and normalize these quantities, see Appendix B.

Indoor air quality is an important dimension not represented in the 3C Reward. An additional reward term can be added to account for it during training, and an air quality constraint can also be used at deployment time. Appendix B further quantifies air quality as a potential reward term or constraint, informed by the ANSI/ASHRAE Standard 62.1 standard for indoor air quality ASHRAE (2022).

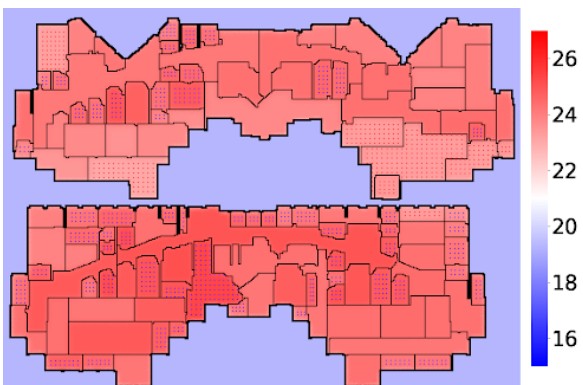

Figure 1: Visualization of an Environment. Colder temperatures are blue; warmer ones are red. Blue and red dots inside the building indicate diffusers dispensing cold and warm air, respectively.

## 4 THE SMART BUILDINGS DATASET

The real-world data and interactive simulated data have the same format. Data are provided as a series of observations, actions, and rewards. The real-world data are in the form of static historical episodes, where actions follow the baseline policy in the building. The simulator and PINN are interactive RL environments where actions can be taken in real-time. The data fall into the following categories:

1. **Environment Data** For each building environment, the dataset contains information on all zones and devices. This includes the name and size of each zone, and the zone, location, sensors, and setpoints of each device.
2. **Observation Data** Observations consist of measurements from all devices in the building (VAV's zone air temperature, gas meter's flow rate, etc.), provided at each timestep.
3. **Action Data** The device setpoint values that the agent wants to set, provided at each timestep
4. **Reward Data** Information used to calculate the reward, as expressed in cost in dollars, carbon footprint, and comfort level of occupants, provided at each timestep

Table 1: Building Information

| BUILDING | FT$^2$ | FLOORS | DEVICES | LOCATION | DURATION |
|---|---|---|---|---|---|
| SB1 | 93,858 | 2 | 173 | MOUNTAIN VIEW, CA | 4 YEARS |
| SB2 | 62,613 | 1 | 144 | MOUNTAIN VIEW, CA | 4 YEARS |
| SB3 | 118,086 | 3 | 281 | MOUNTAIN VIEW, CA | 4 YEARS |
| SB4 | 50,852 | 6 | 128 | SYRACUSE, NY | 2 YEARS |
| SB5 | 5,500 | 2 | 11 | SYRACUSE, NY | 2 YEARS |
| SB6 | 5,120 | 1 | 6 | SYRACUSE, NY | 2 YEARS |
| SB7 | 76,000 | 5 | 148 | NEEDHAM, MA | 1 YEAR |
| SB8 | 74,631 | 5 | 119 | NEEDHAM, MA | 1 YEAR |
| SB9 | 90,650 | 5 | 136 | NEEDHAM, MA | 1 YEAR |
| SB10 | 86,150 | 4 | 231 | NEEDHAM, MA | 1 YEAR |
| SB11 | 58,217 | 4 | 101 | NEEDHAM, MA | 1 YEAR |

The data is from 11 buildings across North America, spans 3 building management systems, and was collected over a 6-year period. This diversity makes it ideal for building environment models that scale. For information on the data format and how to access the benchmark, see Appendix C.

**Data Visualization** We also present a data visualization module for both viewing the real-world data and visualizing the state of the simulator, as shown in Figure 1. Given an observation of a building environment, our visualization module renders a two-dimensional heatmap view of the building. This greatly aids in understanding the data and analyzing how a particular policy is behaving.

## 5 THE SMART BUILDINGS SIMULATOR

Our goal is to apply RL at scale to commercial buildings. For this to be feasible, we must have a lightweight, easy to configure and calibrate simulated environment to train the agent, with high enough fidelity to train an improved control agent. To meet these desiderata, we designed a lightweight simulator based on finite differences approximation of heat exchange, building upon earlier work (Goldfeder & Sipple, 2023). It utilizes tensor operations to enable GPU acceleration with Tensorflow. We proposed a simple automated procedure to go from building floor plans to a custom simulator in a short time, and we designed a calibration and evaluation pipeline, to use data to fine tune the simulation to better match the real world. What follows is a description of our implementation. For details regarding design considerations, see Appendix D.

**Thermal Model for the Simulation** As a template for developing simulators that represent target buildings, we start with a general-purpose high-level thermal model for simulating office buildings, illustrated in Figure 2. In this thermal cycle, we highlight significant energy consumers as follows. The boiler burns natural gas to heat the water, $\dot{Q}_b$ . Water pumps consume electricity $\dot{W}_{b,p}$ to circulate heating water through the VAVs. The air handler fans consume electricity $\dot{W}_{b,in}$ , $\dot{W}_{b,out}$ to circulate the air through the VAVs. A motor drives the chiller's compressor to operate a refrigeration cycle, consuming electricity $\dot{W}_c$ . In some buildings coolant is circulated through the air handlers with pumps that consume electricity, $\dot{W}_{c,p}$.

We selected **hot water supply temperature** $\hat{T}_b$ and **air handler supply temperature** $\hat{T}_s$ as agent actions because they affect the balance of electricity and natural gas consumption, multiple device interactions, and occupant comfort. Greater efficiencies can be achieved with these setpoints by choosing the ideal times and values to warm up and cool down the building in the workday mornings and evenings. Further tradeoffs include balancing the thermal load between hot water heating with natural gas and supply air heating with electricity using the AC or heat pump units.

**Finite Differences Approximation** The diffusion of thermal energy in time and space of the building can be approximated using the method of Finite Differences (FD)(Sparrow, 1993; Lomax et al., 2002), and applying an energy balance. This method divides each floor of the building into a grid of three-dimensional control volumes (CVs) and applies thermal diffusion equations to estimate the temperature of each CV. By assuming each floor is adiabatically isolated, we can simplify the three spatial dimensions into a spatial two-dimensional heat transfer problem. Each CV is a narrow volume bounded horizontally, parameterized by $\Delta x^2$, and vertically by the floor height. The energy balance, shown below, is applied to each discrete CV in the FD grid and consists of the following components: (a) the thermal exchange across each face of the four participating neighbor CVs via conduction or convection $Q_1, Q_2, Q_3, Q_4$, (b) the change in internal energy over time in the CV $Mc\frac{\Delta T}{\Delta t}$, and (c) an external energy source that enables applying local thermal energy from the HVAC model only for those CVs that include an airflow diffuser, $Q_{ext}$. The equation is $Q_{ext} + Q_1 + Q_2 + Q_3 + Q_4 = Mc\frac{\Delta T}{\Delta t}$, where $M$ is the mass and $c$ is the heat capacity of the CV, $\Delta T$ is the temperature change, and $\Delta t$ is the timestep interval.

The thermal exchange in (a) is calculated using Fourier's law of steady conduction in the interior CVs (walls and interior air), parameterized by the conductivity of the CV, and the exchange across the exterior faces of CVs are calculated using the forced convection equation, parameterized by the convection coefficient, which approximates winds and currents surrounding the building. The change in internal energy (b) is parameterized by the density, and heat capacity of the CV. Finally, the thermal energy associated with the VAV (c) is equally distributed to all associated CVs that have a diffuser. Thermal diffusion within the building is mainly accomplished via forced or natural convection currents, which can be notoriously difficult to estimate accurately. We note that heat transfer

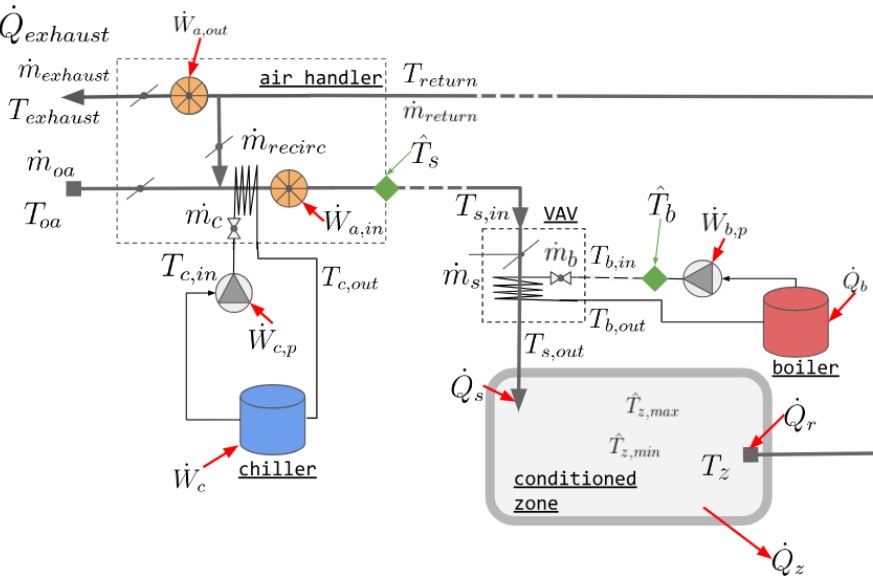

Figure 2: Thermal model for simulation. A building consists of conditioned zones, where the mean air temperature of the zone $T_z$ should be within upper and lower setpoints, $\hat{T}_{z,max}$ and $\hat{T}_{z,min}$. Thermal power for heating or cooling the room is supplied to each zone, $\dot{Q}_s$, and recirculated from the zone, $\dot{Q}_r$ from the HVAC system, with additional thermal exchange $\dot{Q}_z$ from walls, doors, etc. The AHU supplies the building with air at supply air temperature setpoint $\hat{T}_s$ drawing fresh outside air, $\dot{m}_{OA}$, at temperatures, $T_{OA}$, and returning exhaust air $\dot{m}_{exhaust}$ at temperature $T_{exhaust}$ to the outside using intake and exhaust fans, $\dot{W}_{a,in}$ and $\dot{W}_{a,out}$. A fraction of the return air can be recirculated, $\dot{m}_{recirc}$. Central air conditioning is achieved with a chiller and pump that join a refrigeration cycle to the supply air, consuming electrical energy for the AC compressor $\dot{W}_c$ and coolant circulation, $\dot{W}_{c,p}$. The hot water cycle consists of a boiler that maintains the supply water temperature at $T_b$ heated by natural gas power $\dot{Q}_b$, and a pump that circulates hot water through the building, with electrical power $\dot{W}_{b,p}$. Supply air is delivered to zones through VAVs.

using air circulation is effectively the exchange of air mass between CVs, which we approximate by a randomized shuffling of air within thermal zones, parameterized by a shuffle probability and radius. For further details see Appendix E.

**Simulator Configuration** For RL to scale to many buildings, it is critical to be able to easily and rapidly configure the simulator to any arbitrary building. We designed a procedure that, given floor-plans and HVAC layout information, enables generating a fully specified simulation very rapidly. For example, on SB1, consisting of 2 floors and 173 devices, a single technician was able to configure the simulator in under 3 hours. Details of this procedure are in Appendix F.

**Simulator Calibration and Evaluation** In order to calibrate the simulator using sensor data, we must have a metric with which to evaluate our simulator's fidelity, and an optimization method to improve our simulator on this metric.

**$N$-Step Evaluation** We propose a novel evaluation procedure based on $N$-step prediction. Each iteration of our simulator is designed to represent a five-minute interval. To evaluate the simulator, we take a chunk of real data, consisting of $N$ consecutive observations. We then initialize the simulator so that its initial state matches that of the starting observation and run the simulator for $N$ steps, replaying the same HVAC policy as was used in the real world. We then calculate our simulation fidelity metric, which is the mean absolute error of the temperatures in each temperature sensor at each timestep, averaged over time. More formally, we define the Temporal Spatial Mean Absolute Error (TS-MAE) of $Z$ zones over $N$ timesteps as:

$$\epsilon = \sum_{t=1}^{N} \frac{1}{N} \left[ \frac{1}{Z} \sum_{z=1}^{Z} |T_{real,t,z} - T_{sim,t,z}| \right] \tag{1}$$

Where $T_{real,t,z}$ is measured zone air temperature for zone $z$ at timestamp $t$, and $T_{sim,t,z} = \frac{1}{|C_z|} \sum_{c=1}^{C_z} T_{t,c}$ is mean temperature of all CVs $C_z$ in zone $z$ at time $t$.

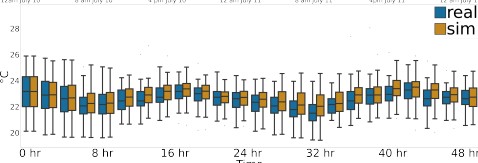
Figure 3: Drift Over 48 hrs on Train Set

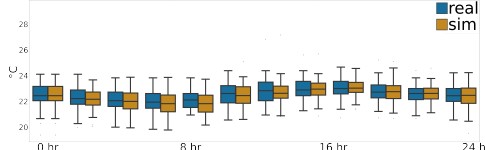
Figure 4: Drift Over 24 hrs on Validation Set

**Hyperparameter Calibration** Once we define our simulation fidelity metric, TS-MAE, we can minimize this error by hyperparameter tuning several physical constants and other variables using black-box optimization methods. We chose the method outlined in Golovin et al. (2017), which automatically chooses the most appropriate strategy from a variety of popular algorithms.

## 5.1 SIMULATOR CALIBRATION DEMONSTRATION

We now provide an example of our calibration procedure to tune the physical parameters and minimize prediction error.

**Setup** We configured the simulator to match SB1, with two stories, a combined surface area of 93,858 square feet, and 170 HVAC devices. Using the configuration pipeline, we went from floor plan blueprints to a fully configured simulator, a process that took a single technician less than three hours to complete. To calibrate, we took SB1 data from 3 days, from Monday July 10, 2023 12:00 AM, to Thursday July 13, 2023 12:00 AM. The first 2 days were used as a train set, and the third day as validation, as can be seen in Table 2. All times are local to the building.

**Calibration Procedure** We ran hyperparameter tuning for 4000 iterations to optimize the TS-MAE, as outlined in equation 1, on the training data. We reviewed the physical constants that yielded the lowest simulation error from calibration. Densities, heat capacities, and conductivities plausibly matched common interior and exterior building materials. However, the external convection coefficient is higher than under the weather conditions and likely is compensating for the radiative losses and gains, which were not directly simulated. For details about the hyperparameter tuning procedure, including the parameters varied, the ranges given, and the values found, see Appendix G.

**Calibration Results** Table 2 shows the predictive results of our calibrated simulator, on $N$-step prediction, for the train scenario, where $N = 576$, representing a two-day predictive window, and the test scenario, where $N = 288$, representing a one day window. We calculate the TS-MAE as defined in equation 1. We show results for the hyperparameters that best fit the train set, as well as for an uncalibrated simulator as a baseline. The validation data was never provided to the tuning process. Our tuning procedure drifts only $0.56°C$ on average over a 24-hour validation period.

**Visualizing Temperature Drift Over Time** Figure 3 illustrates temperature drift over time for the training scenario. At each timestep, we calculate the spatial temperature for all sensors in both the real building and simulator and present them

| SPLIT | START | END | CALIB. $\epsilon$ | UNCALIB. $\epsilon$ |
|-------|-------|-----|-------------------|---------------------|
| TRAIN | 23-07-10 | 23-07-12 | 0.717 $°C$ | 1.971 $°C$ |
| VAL. | 23-07-12 | 23-07-13 | 0.566 $°C$ | 1.618 $°C$ |

Table 2: Training and test data scenarios

as side-by-side boxplot distributions for comparison. Figure 4 shows the same for the validation scenario. Here we can see that our simulator temperature distribution maintains a minimal drift from the real world, although it does seem less reactive to daily fluctuation patterns, which may be due to the lack of a radiative heat transfer model.

**Visualizing Spatial Errors** Figure 5 illustrates the results of this predictive process over a 24-hour period, on the validation data. It displays a heatmap of the spatial temperature difference throughout the building, between the real world and simulator, after 24 hours of the simulator making predictions. The ring of blue around the building indicates that our simulator is too cold on the

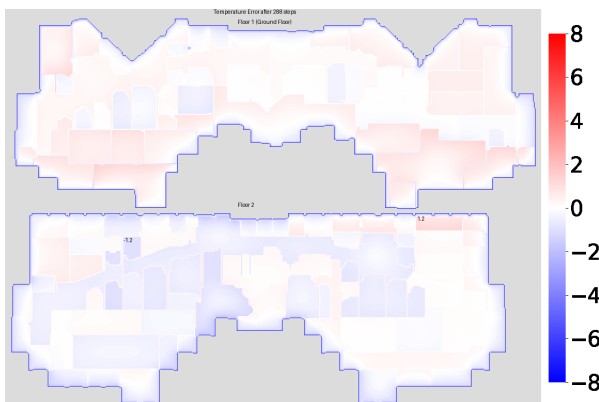

Figure 5: Visualization of simulator drift after 24 hours on validation data. A heat map represents the temperature difference between the simulator and the real world, with red indicating the simulator is hotter, blue indicating it is colder, and white indicating no difference. The zones with the max and min differences are indicated by displaying the difference above them.

perimeter, which implies that the heat exchange with the outside is happening more rapidly than it would in the real world. The inside of the building, at least on the first floor, contains significant amounts of red, indicating that despite the simulator perimeter being cooler than the real world, the inside is warmer. This implies that our thermal exchange within the building is not as rapid as that of the real world. We suspect that this may be because our simulator does not have a radiative heat transfer model. Lastly, there is a large amount of white in this image, indicating that for the most part, even after 24 hours of making predictions on the validation data, our calibration process was successful and the fidelity remains high. For more visuals of spatial errors, see Appendix H.

## 6 THE SMART BUILDINGS PINN MODEL

**Problem Formulation** Recent work developed a modularized neural network model incorporating physical priors for building energy modeling, where different modules were designed to estimate distinct heat transfer terms in the dynamic building system (Jiang & Dong, 2024). Building on this foundation, we updated the model structure to focus on control optimization. We consider the modeling task as a discrete-time dynamic system formulated in a state-space representation, with state variables, control inputs, and disturbance variables. As a control-oriented model, the primary focus is on managing model complexity while ensuring that its responses align with physical laws. To balance complexity and accuracy, we simplified the heat transfer terms by dividing them into three components: HVAC, adjacent zone, and other disturbances (e.g., outside air temperature, solar radiation, occupancy, and time features represented as sinusoidal functions). We further incorporated physical consistency constraints. The energy balance is expressed as:

$$x(t+1) = x(t) + \frac{1}{cM}\Delta Q = x(t) + f_{NN_A}\Big(f_{NN_B}(u(t))$$

$$+ f_{NN_E}(x(t), w(t)) + f_{NN_{\text{adj}}}(x(t))\Big) \quad (2)$$

Where $x$ represents the state variable (space air temperature), $M$ is the mass, and $c$ is the heat capacity of the space. $\Delta Q$ denotes the energy change within one timestep, which includes HVAC input and other heat transfer terms from conduction, convection, and radiation. $f_{NN_A}, f_{NN_B}, f_{NN_E}$, and $f_{NN_{\text{adj}}}$ are separate neural network modules to learn heat transfer dynamics, respectively.

**Modular Design** We develop an encoder-decoder structure for each neural network module. The encoder captures the thermal initial impact, providing a stable latent hidden state. Subsequently, we take a measurement of the current state to correct prediction errors at each timestep. The decoder then predicts outputs based on disturbance variables and control inputs. Beyond that, physical consistency is enforced through hard model parameter constraints to ensure that the control gain is positive. For example, additional cooling should decrease the space air temperature. This constraint, shown below, ensures that the partial derivative of the state variables to its control input is always positive before the current timestep $\frac{\partial x_t}{\partial u_k} > 0$ for $k < t$.

**From Single-zone to Multi-zone** To extend our model from a single-zone to a multi-zone framework, we integrated an adjacent heat transfer module to calculate conduction heat transfer based on

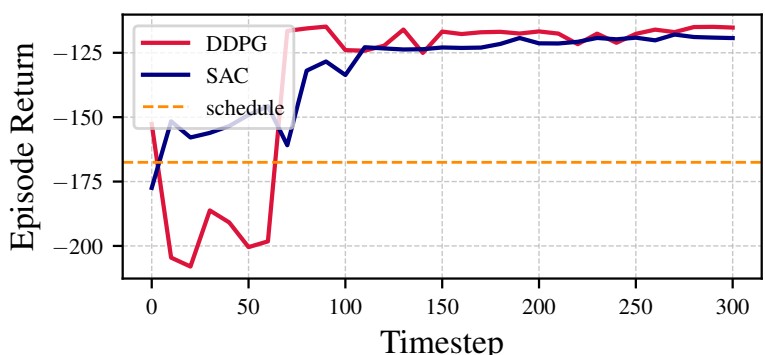

Figure 6: SAC and DDPG returns

temperature differences between zones. A skew-symmetric matrix, developed from the graph adjacency matrix, was used to represent this heat transfer term, ensuring that the heat transfer between two adjacent zones is equal in magnitude but opposite in direction.

**Model Training** The dataset used to train the model in these experiments was collected from SB4 between January 2023 and August 2024 and was divided into training, validation, and testing sets in a 7:2:1 ratio. All features were normalized using a Min-Max Scaler, mapping the values to the range $[-1, 1]$. To emulate a real-world training process, the size of the training data was gradually increased, starting with seven days and extending to half a year, reflecting the progressive availability of data over time. To prevent overfitting, k-fold cross-validation and early stopping were applied.

**Model Validation** A well-predicting model does not necessarily ensure correct dynamic responses. To evaluate model performance, we use three metrics: 1) Mean Absolute Error (MAE): A standard metric to assess model accuracy. 2) Temperature Response Violation, where at each timestep we introduce varying levels of HVAC input from -4 kW to 4 kW to perform a sanity check. If the temperature increases with additional cooling or vice versa, it is considered a temperature violation. 3) Maximum mean discrepancy (MMD): MMD quantifies the difference between two sets of samples by taking the maximum difference in sample averages over a kernel function. MMD evaluates the similarity between the model's responses and ground truth responses collected from measured data.

## 7 LEARNING AN IMPROVED CONTROL POLICY

Here we provide an example of training an RL Agent on the simulator to generate an improved policy over the current rule-based baseline programmed by the building operators, using building SB1. We demo our benchmark using SAC and DDPG, and compare the learned policy with the baseline policy currently used in the real building. Both actor and critic are feedforward networks. We ran hyperparameter tuning, again using the method from Golovin et. al. (Golovin et al., 2017), to choose the dimensionality of the critic network and actor network, the batch size, the critic learning rate and actor learning rate, and $\gamma$. The training episode lasted 14 days, and returns can be seen in Figure 6. For details and a performance comparison between the learned policy and baseline, with a breakdown on setpoint deviation, carbon emissions, electricity, and natural gas, see Appendix J.

## 8 LIMITATIONS AND CONCLUSION

A current limitation is that our simulator requires additional physical model enhancements. While our benchmark is grounded in real data, we do not have results on training a model and deploying it on these buildings, something we leave for future work. We present a high-quality interactive HVAC Control Suite, with an explicit focus on solutions that transfer. Our benchmark has three parts, representing the state of the art of open source HVAC data, scalable data-driven simulation, and physically informed dynamics models. We believe this benchmark will facilitate collaboration, reproducibility, and progress, making an important contribution towards sustainability.

## 9 ETHICS STATEMENT

Care was taken to ensure that our data release does not harm an individuals or entities. Any sensitive data, such as building location, or location of specific desks, has been fully redacted.

## 10 REPRODUCIBILITY STATEMENT

All of our work and data is open source and easily reproducible. While a link to github was not included, in order to ensure anonymity, our code was included in supplementary materials.

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

# A    LEARNING AN OPTIMAL CONTROL POLICY

**Reinforcement Learning** (RL) is a branch of machine learning that attempts to train an agent to choose the best actions to maximize the expected long-term, cumulative reward (Sutton & Barto, 2018). The set of parameters $\boldsymbol{\theta}^*$ of the optimal policy can be expressed as:

$$\boldsymbol{\theta}^* = \arg\max_{\theta} \mathbb{E}_{\tau \sim \pi_\theta(\tau)} \left[ \sum_t \gamma^t R(S_t, A_t) \right]$$

where $\theta$ is the current policy parameter, and $\tau$ is a trajectory of states, actions, and rewards over sequential timesteps $t$. Over time, the agent explores the action space and learns to maximize the reward over the long term for each given state. A discount factor $\gamma$ reduces the value of future rewards amplifying the value of the near-term reward. When this cycle is repeated over multiple episodes, the agent converges on a state-action policy that maximizes the long-term reward. To converge to the optimal policy, the agent requires many iterations to explore the policy space, making online training directly on the real-world building inefficient, dangerous and impractical. Therefore, it is necessary to enable offline learning, where the agent can train in an efficient sandbox environment that adequately emulates the dynamics of the building before being deployed to the real world.

When applying RL to find an optimal policy in a complex dynamical environment such as a building, there are generally two possible approaches. The *model-free RL* approach involves learning a policy by directly following gradient signal from the reward function, similar to error backpropagation from a loss function. This is a seemingly straightforward approach but care must be taken to accommodate the stochastic nature of the MDP and the environment, as well as other sources of noise such as sensor noise and delays. A variety of statistical smoothing techniques are often used, generally leading to a slow convergence rate. This process can sometimes get stuck in local optima, oscillate in cycles, or be too slow to keep up with a changing environment.

The alternative *model-based RL* process involves learning an internal model (or making use of an existing one, such as our simulator or PINN) that predicts the likely state that will result from certain state+action combinations. This internal model (or models) can then be used by an optimizer/learner to explore potential policies in deeper and more sophisticated ways, sometimes even using multi-step look-ahead and other heuristic search schemes suitable for highly rugged reward landscapes or environments with very long-term nonlinear rewards. The learned internal model can be simple, based only on current state and action, or deep, in that it will consider long and short term history and other factors such as weather predictions provided by other models. Model based RL can potentially make better use of past experiences, but in turn requires more computation and could potentially converge prematurely on wrong strategies. Note that both *model-based* and *mode-free* RL must allow for some off-policy exploration in order to learn the landscape, leading to the *exploration-exploitation* dilemma.

Ideally, the internal model used by model-based RL systems – like pre-trained generative systems in general – can contain knowledge that is potentially transferable across tasks. For example, once an internal model can predict the state resulting from certain actions, it can be used for a variety of tasks, such as heating, cooling, minimizing boiler fluctuations, or any arbitrary objective that can be calculated on a trajectory. A dynamics model can also be used to optimize non-RL policies, such as heuristic rule tables, decision trees, or even conventional PID controllers. We view this kind of classical *model-predictive control* (MPC) as a baseline, as we hope RL agents will learn to outperform them.

# B    REWARD FUNCTION DETAILS

We call our reward function the 3C Reward, because it is made up of a combination of three factors: Comfort, Cost, and Carbon. The purpose of the reward function is to provide the agent with a feedback signal after each action about the quality of the current and past actions performed. We combine the different objectives as a normalized, weighted sum of maintaining comfort conditions, electrical cost, and carbon cost:

$$R_{3C} = u \times C_1 + v \times C_2 + w \times C_3$$

where $C_1$ represents normalized comfort conditions, $C_2$ normalized energy cost and $C_3$ normalized carbon emission. Constants $u$, $v$, $w$ represent operator preferences, allowing them to weigh the relative importance of cost, comfort and carbon consumption.

Each value $C_1, C_2, C_3$, is bounded by the range $[-1, 0]$, where worst performance is $-1$ and the ideal performance upper-bound is $0$ Thus the reward function in an aggregate is formulated as an approximate regret function, bounded in the range [-1,0], and represents an offset from the best case where comfort conditions are perfectly maintained, without consuming energy and emitting carbon. Each of the sub functions $C_1, C_2, C_3$ will be elaborated next.

## B.1 COMFORT LOSS FUNCTION ($C_1$)

Besides zone air temperature, other factors such as ventilation, drafts, solar exposure, humidity and air quality affect human comfort and productivity in office buildings. However, for now we are focused solely on temperature as the indicator of the comfort level in the office buildings. As additional sensors are deployed and the other factors are measured, they should be considered in the definition of an enhanced comfort loss function.

Studies have shown that a relationship exists between work performance and temperature and air quality (Deng et al., 2024). For example, in Seppanen et al. (2006), work performance was quantified as the mean time required to complete common office tasks (e.g., text processing, bookkeeping calculations, telephone customer service calls, etc.). Performance was shown to increase gradually with temperatures increasing up to 21-22°C and decreasing at temperatures beyond 23-24°C. Therefore, when temperatures deviate outside setpoints, the comfort loss should also be smooth and monotonically increasing.

Thus, the following rules were selected to govern the comfort loss function:

1. Setpoints define the comfort standards, and no penalty should be applied whenever the zone temperature is within heating and cooling setpoints.

2. Comfort is undefined when the zone is unoccupied: if the zone is unoccupied, comfort loss is zero, regardless of zone temperature.

3. Comfort decays smoothly and monotonically as the temperatures drift from setpoints, and occupants are tolerant to small setpoint deviations. Therefore, small setpoint deviations should have a small comfort penalty, and the penalty should smoothly increase as the deviations increase.

4. Large setpoint deviations should approach a maximum, bounded penalty, where a zone becomes completely intolerable for its occupants.

The comfort loss function represents a bounded penalty term for occupied zones that have zone air temperatures outside of setpoint and covers three adjacent temperature intervals: below cooling setpoint $T_z < \hat{T}_{heating}$, inside setpoints $\hat{T}_{heating} \leq T_z \leq \hat{T}_{cooling}$, and above cooling setpoint $\hat{T}_{cooling} < T_z$

We propose a logistic sigmoid parameterized by $\lambda$ and $\Delta$ to represent the smooth decay (increase loss) of comfort below the heating and above the cooling setpoints. Parameter $\lambda$ is a stiffness coefficient that affects the slope of the decay and parameter $\Delta$ represents the offset in $°C$ from the set point where halfway loss value (0.5) occurs. Additionally we define a step function $\delta(k) = 1$ when the zone has at least one occupant $(k > 0)$, and $\delta(k) = 0$ otherwise.

$$h_z(T_z, k_z, \hat{T}_{heating}, \hat{T}_{cooling}) = \begin{cases} \frac{\delta(k_z)}{1+e^{-\lambda(T_z - \hat{T}_{heating} + \Delta)}} - 1 & T_z < \hat{T}_{heating}] \\ 0 & \hat{T}_{heating} \leq T_z \leq \hat{T}_{cooling} \\ \frac{-\delta(k_z)}{1+e^{-\lambda(T_z - \hat{T}_{cooling} - \Delta)}} & \hat{T}_{cooling} < T_z \end{cases}$$

The chart below shows the comfort loss curve with common setpoints, where the horizontal axis represents zone air temperature and the vertical axis represents the loss. The heating and cooling setpoints were taken from data recordings.

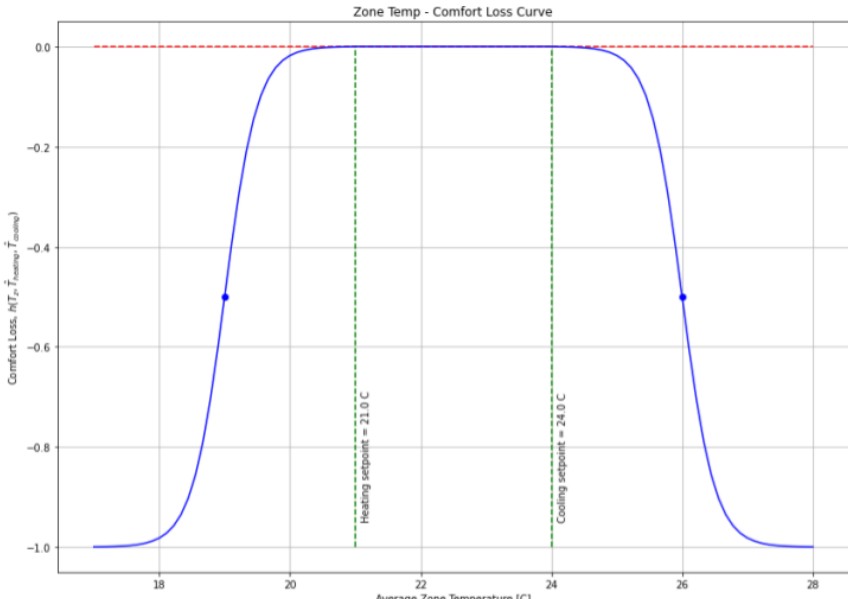

Figure 7: Setpoint Diagram

Finally, we compute the average of all zone comfort losses as the building's overall comfort loss:

$$h_t(S_t) = \frac{1}{|Z|} \times \sum_{z \in Z} h_z(T_z, k, \hat{T}_{heating}, \hat{T}_{cooling})$$

**Live Occupant Feedback** The idea of human feedback shaping the agent's policy may be particularly suitable for the smart buildings project and has been detailed in Knox and Stone 2009. While not implemented in the initial version of the reward function, the comfort loss function can be extended with an occupant feedback signal reflecting discomfort (e.g., "too hot" or "too cold") in a variety of methods like Mozer 1998 (Mozer, 1998). The agent's goal should be to minimize this type of feedback, and the regret should be increased anytime this feedback signal is received. Suppose one or more occupants in zone $z$, provided a "too cold" feedback signal, $\hat{T}_{heating}$ may be increased by a small amount from the baseline setpoint configuration, and may smoothly return to the baseline smoothly after an appropriate delay.

**Stochastic Occupancy Model** The occupancy signal $k_z$ is the average number of occupants in zone $z$ during a timestep $t_i - t_{i-1}$ and is used in computing the comfort loss function described above. Ideally, the occupancy signal is obtained from motion detection sensors or secondary indicators of occupancy, such as wifi signals, badge swipes, calendar appointments, etc. However, a data-driven occupancy signal was not available for the initial dataset, and the following stochastic occupancy model is used instead.

For workdays, we would like model occupancy as a process in the zone where a max number of occupants, $k_{z,max}$ arrive at random times in an arrival window $[\tau_{in,start}, \tau_{in,end}]$, and depart the zone in a departure window $[\tau_{out,start}, \tau_{out,end}]$. The arrivals and departures should occur evenly within the intervals and the expectation of the arrival time should be at the halfway point of the arrival interval:

$\mathbb{E}[\text{occupant arrival time}] = \frac{1}{2}(\tau_{in,end} - \tau_{in,start}) + \tau_{in,start}$

Likewise, the expectation of the departure time should be at the halfway point of the departure interval:

$$\mathbb{E}[\text{occupant departure time}] = \tfrac{1}{2}(\tau_{out,end} - \tau_{out,start}) + \tau_{out,start}$$

If the number of timesteps within the arrival and departure intervals is $n_{arrival}$ and $n_{departure}$, this process can be modeled as a geometric distribution where each timestep and occupant is a Bernoulli trial with probabilities:

$P(\text{ occupant arrives — occupant has not yet arrived }) = \frac{2}{n_{arrival}}$ and $P(\text{ occupant departs — occupant has arrived }) = \frac{2}{n_{departure}}$ During holidays and weekends, the zones are not occupied: $k_z = 0$.

## B.2 ENERGY COST FUNCTION ($C_2$)

The energy cost function $C_1(S_t)$ is a normalized, aggregate cost estimate from consuming electrical and natural gas energy during one timestep. The cost function is the ratio of the actual energy used to the maximum energy capacity that ranges between 0: no cost incurred; and 1: maximum cost incurred.

$$C_2(S_t) = -\frac{actual\ energy\ cost}{cost\ at\ max\ energy\ capacity}$$

General energy cost can be calculated as the product of the mean power applied, the time interval, and the cost per unit energy at the time of the interval, where we use $W, \dot{W}$ to represent electrical/mechanical energy, power, and $Q, \dot{Q}$ to represent thermal energy and power from natural gas. Since all four terms contain the same interval $t_i - t_{i-1}$, they cancel out, allowing us to use power instead of energy. As described above, pumps, blowers, and AC/refrigeration cycles consume electricity and water heaters/boilers consume natural gas. Therefore the total energy and cost is the sum of each energy consumer cost used over the interval:

$$C_2(S_t) = -\frac{(\dot{W}_a + \dot{W}_m + \dot{W}_p) \times p_e(t) + \dot{Q}_g \times p_g(t)}{(\dot{W}_{a,max} + \dot{W}_{m,max} + \dot{W}_{p,max}) \times p_e(t) + \dot{Q}_{g,max} \times p_g(t)}$$

Where $\dot{W}_a$ and $\dot{W}_{a,max}$ are the actual and max electrical power for the AC/refrigeration cycle, $\dot{W}_m$ and $\dot{W}_{m,max}$ are the actual and max electrical power for the blowers/air circulation, $\dot{W}_p$ and $\dot{W}_{p,max}$ are the actual and max pump electrical power, and $\dot{Q}_g$ and $\dot{Q}_{g,max}$ are the actual and max thermal power . Terms $p_e(t)$ and $p_g(t)$ are the electricity and gas price per energy incurred over the interval at time $t$.

The actual power terms in the numerator are estimated from the device observations and the device's fixed parameters using standard HVAC energy conversions. The max power terms in the denominator are derived from device ratings, which define the maximum operating nouns of the device.

## B.3 CARBON EMISSION COST FUNCTION ($C_3$)

Similar to the energy cost function, carbon emission cost function is a function of the electrical and natural gas power used during the interval. The carbon emission cost function $C_3$ is a normalized, aggregate cost estimate from the emission of carbon mass by consuming electrical and natural gas energy during one timestep. The cost function is the ratio of the actual carbon used to the maximum carbon emitted that ranges between 0: no emission cost incurred; and 1: maximum emission cost incurred.

$$C_3(S_t) = -\frac{actual\ carbon\ mass\ emitted}{maximum\ carbon\ emitted}$$

The carbon emission cost is similar to the energy cost function described above, except that we replace the price terms $p_e, p_g$ with emission terms $r_e, r_g$ that convert the power to carbon emission rates.

While the emission rate for natural gas is fairly constant, the emission rate for electricity is dependent on the utility's current renewable energy supply and consumer load during the interval and may fluctuate significantly.

### B.4 IMMEDIATE AND DELAYED REWARD RESPONSES

The reward function is a weighted average of maintaining temperature setpoints in occupied zones, while minimizing energy cost, and minimizing carbon emission. Both energy and carbon emission cost functions provide a low latency response, because actions have an almost immediate effect on the reward. For example, lowering the supply water temperature setpoint will reduce the flow of natural gas to the burner, bringing $\dot{Q}$ down in the next step. However, the effect of increasing water temperature on the comfort loss function may be delayed by multiple timesteps, due to the thermal latency in the building. This thermal latency is due to inherent heat capacity and thermal resistance within the building that has a dampening effect on diffusing heat throughout the building. This means that some settings of $u, v, w$ may cause undesirable effects. Experiments with the simulation indicate that too strong weights (e.g., $u + v \geq 0.6$) toward energy cost and/or carbon emission may lead the agent to lower the water temperature, which can cause the VAVs to increase their airflow demand to compensate for a lower supply air temperature, since thermal energy flow is a tradeoff between air mass flow and water heating at the VAV's heat exchanger. Consequently, the increased airflow demand results in a much higher, delayed electrical energy consumption by the blowers to meet the zone airflow demand.

### B.5 INDOOR AIR QUALITY

Indoor air quality is important for multiple reasons, from mitigating exposure to airborne viruses to managing humidity. While the action space in this paper focuses on the hot water and air handler supply temperatures, the amount of fresh air being brought into the air handler is an important factor for air quality. Most air handler units will have a minimum amount of fresh air as specified by a minimum outside air damper position, and the damper position is dependent on the supply air temperature setpoint. Nevertheless, when deploying air handler policies, it is important to verify that the building is receiving sufficient fresh air.

The ANSI/ASHRAE Standard 62.1 standard for indoor air quality ASHRAE (2022) recommends the following equation for minimum outside airflow:

$$V_{min} = R_p \cdot P_z + R_a \cdot A_z \tag{3}$$

Where $R_p$ is the rate per person, $P_z$ is the zone population, $R_a$ is the rate per area, and $A_z$ is the zone area. $R_p$ for office spaces is 5 CFM per person, but other types of spaces may require higher rates of 10-20 CFM per person, or 30 CFM per person for infection risk management as specified by ASHRAE Standard 241-2023 Sherman & Jones (2023). The $R_a$ for office spaces is 0.06 CFM per square foot, but this could similarly be increased for different space types.

There are two ways to incorporate this standard: one is by adding a reward term and the other is by applying a constraint at deployment time, both described below.

**Modified Reward and Action Space** An air quality term could be added to the 3C reward to penalize states with lower outside airflow. Similar to the comfort reward term, this could use a logistic sigmoid to represent a smooth, bounded decay, with 0.5 loss being the point where outside airflow $V_{OA}$ equals the minimum standard $V_{min}$ as defined in equation 3.

$$R(V_{OA}) = \frac{1}{1 + e^{-\lambda(V_{OA} - V_{min})}} \tag{4}$$

Parameter $\lambda$ is a stiffness coefficient that affects the slope of decay, and might have a value between 1 and 10 depending on the requirements of the deployment space.

To better optimize the overall system, the static pressure setpoint and outside air damper position could be added to the action space. Another way to incorporate air quality would be to add a feasibility constraint instead of a reward term during policy learning (Chen et al., 2021).

**Constraining at Deployment** When deploying a learned 3C policy, one can calculate the minimum outside airflow constraint (equation 3) using the deployment zone's occupancy and square footage. The outside air damper position can be set to increase when the outside airflow, as measured by static pressure sensors, drops below this constraint. Another way to incorporate indoor air quality would be to combine it with predictive control Wang & Dong (2023).

## C DATASET FORMAT

### C.1 BENCHMARK ACCESS

The data is part of Tensorflow Datasets (TFDS), and can be downloaded from [link redacted for blind review].

The code is available at [link redacted for blind review].

### C.2 DATASET FORMAT

Here, we will elaborate on the exact format of the dataset.

Having applied the RL paradigm, the data in our dataset falls under the following categories:

1. **Environment Data** General information about the environment, such as the number of devices and zones, and their names and device types. This is provided once per building environment

2. **Observation Data** The measurements from all devices in the building (VAV's zone air temperature, gas meter's flow rate, etc.), provided at each timestep

3. **Action Data** The device setpoint values that the agent wants to set, provided at each timestep

4. **Reward Data** Information used to calculate the reward, as expressed in energy cost in dollars, carbon emission, and comfort level of occupants, provided at each timestep

### C.3 ENVIRONMENT DATA

This is the data that provides, once per environment, details about the environment such as number of devices or zones. There are four types of data: ZoneInfo, DeviceInfo, Floorplan, and Device Locations.

1. **ZoneInfo:** The `ZoneInfo` defines thermal spaces or zones in the building and provides zone-to-device association, which enables using the associated VAVs' zone air temperatures to estimate the zone's temperature.

2. **DeviceInfo:** The HVAC devices in the building are defined in the `DeviceInfo`. Each device exposes a map of `observable_fields` and `action_fields`. The `observable_fields` represent the observable state of the building in native units, and the `action_fields` are available setpoints exposed by the building that the agent may add to its action space. Currently `observable_fields` and `action_fields` are floating point values, but may be expanded to categorical values in the future.

3. **Floorplan:** This is a 2d matrix, where there are 4 possible values in each cell: outside air, inside air, exterior wall, interior wall.

4. **Device Locations:** This is a dictionary of device names to cells in the floorplan. For each device, the map provides the cells corresponding to the room that the device is located in.

## C.4   OBSERVATION DATA

This includes the measurements from all devices in the building (VAV's zone air temperature, gas meter's flow rate, etc.), provided at each timestep. This data is given as a matrix of TxM, where there are T timesteps and M measurements per timestep. A list of T timestamps is provided, as well as a list of M measurement names, defining the rows and columns of the matrix.

## C.5   ACTION DATA

This consists of the device setpoint values that the agent wants to set, provided at each timestep. This data is given as a matrix of TxS, where there are T timesteps and S setpoints per timestep. A list of T timestamps is provided, as well as a list of S setpoint names, defining the rows and columns of the matrix.

## C.6   REWARD DATA

This includes information used to calculate the reward, as expressed in cost in dollars, carbon footprint, and comfort level of occupants, provided at each timestep The reward data is further divided into two categories:

1. **RewardInfo:** The values that are used as inputs to calculate the reward

2. **RewardResponse:** Containing the scalar reward signal obtained by passing the above functions into our 3C reward function

The building updates the `RewardInfo` at each timestep and provides the reward function necessary inputs to compute the 3C Reward Function. The data contained in the `RewardInfo` is bounded by the step's interval from `start_timestamp` to `end_timestamp` in UTC. The `RewardInfo` has mean energy rate estimates (i.e. power in Watts) that can be treated as constants over the interval. Given the interval and a constant rate value over the interval, the reported power in Watts can be easily converted into energy in kWh. The `RewardInfo` contains maps of three types of specialized data structures:

- The `ZoneRewardInfo` provides information about the zone air temperature measurements, temperature setpoints, airflow rate and setpoint, and average occupancy for the timestep. Each instance is indexed by its unique zone ID.

- The `AirHandlerRewardInfo` describes the combined electrical power in W use of the intake/exhaust blowers, and the electrical power in W of the refrigeration cycle. Since a building may have more than one air handler, the air handler objects are values in a map keyed by the air handlers' device IDs.

- The `BoilerRewardInfo` contains the average electrical power in W used by the pumps to circulate water through the building, and the average natural gas power in W used to heat the water in the boiler. Since there may be more than one hot water cycle in the building, each `ZoneRewardInfo` is placed into a map keyed by the hot water device's ID.

The reward function converts the current `RewardInfo` into the `RewardResponse` for the same interval as the `RewardInfo`. The agent's reward signal is `agent_reward_value`. Since the reward returned to the agent is a function of multiple factors, it is useful for analysis to show the individual components,m such as carbon mass emitted, and the electrical and gas costs for the step.

The data is again stored as matrices, one of RewardInfo and one of RewardResponse. The RewardInfo matrix is dimension TxR, where T is the number of timesteps and R the number of factors used to calculate the reward, again with a list of T timesteps and R ids that determine which device and meter each column refers to. Similarly, the RewardResponse is Dimension TxC, where T is timesteps as before, and C is the number of reward function constants + 1 for the scalar reward itself.

## D  SIMULATOR DESIGN CONSIDERATIONS

A fundamental trade-off when designing a simulator is speed versus fidelity, as depicted in Figure 8. Fidelity is the simulator's ability to reproduce the building's true dynamics that affect the optimization process. Speed refers to both simulator configuration time, i.e., the time required to configure a simulator for a target building, and the agent training time, i.e., the time necessary for the agent to optimize its policy using the simulator.

Every building is unique, due to its physical layout, equipment, and location. Fully customizing a high-fidelity simulation to a specific target building requires nearly exhaustive knowledge of the building structure, materials, location, etc., some of which are unknowable, especially for legacy office buildings. This requires manual "guesstimation" which can erode the accuracy promised by high-fidelity simulation. In general, the configuration time required for high-fidelity simulations limits their utility for deploying RL-based optimization to many buildings. High-fidelity simulations also are affected by computational demand and long execution times.

Alternatively, we propose a fast, low-to-medium-fidelity simulation model that was useful in addressing various design decisions, such as the reward function, the modeling of different algorithms. and for end-to-end testing. The simulation is built on a 2D finite-difference (FD) grid that models thermal diffusion, and a simplified HVAC model that generates or removes heat on special "diffuser" CV in the FD grid. While the uncalibrated simulator is of low-to-medium fidelity, the key additional factor is data. We collect recorded observations from the target building under baseline control, and use that data to **calibrate** the simulator, by adjusting the simulator's physical parameters to minimize difference between real and simulated data. We believe this approach hits the sweet spot in this tradeoff, enabling scalability while maintaining a high enough level of fidelity to train an improved policy.

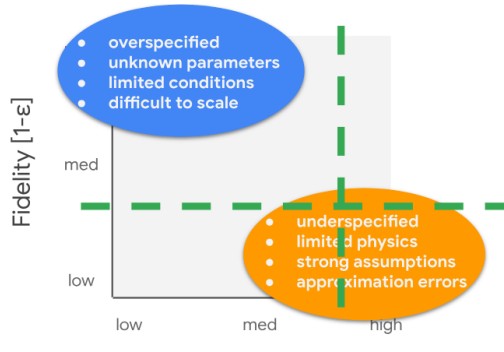

Figure 8: Simulation Fidelity vs. Execution Speed. The ideal operating point for training RL agents for energy and emission efficiency is a tradeoff between fidelity, depicted as 1 minus a normalized error $\epsilon$ between simulation and real, and execution speed, as measured by the number of training steps per second. Additional consideration also includes the time to configure a custom simulator for the target building. While many approaches tend to favor high fidelity over execution, speed, our approach argues a low-to-medium fidelity that has a medium-to-high speed is most suitable for training an RL agent.

Thus, a simulator models the physical system dynamics of the building, devices, and external weather conditions, and can train the control agent interactively, if the following desiderata are achieved:

1. The simulation must produce the same observation dimensionality as the actual real building. In other words, each device-measurement present in the real building must also be present in the simulation.

2. The simulation must accept the same actions (device-setpoints) as the real building.

3. The simulation must return the reward input data described above (zone air temperatures, energy use, and carbon emission).

4. The simulation must propagate, estimate, and compute the thermal dynamics of the actual real building and generate a state update at each timestep.

5. The simulation must model the dynamics of the HVAC system in the building, including thermostat response, setpoints, boiler, air conditioning, water circulation, and air circulation. This includes altering the HVAC model in response to a setpoint change in an action request.

6. The time required to recalculate a timestep must be short enough to train a viable agent in a reasonable amount of time. For example, if a new agent should be trained in under three days (259,200 seconds), requiring 500,000 steps, the average time required to update the building should be 0.5 seconds or less.

7. The simulator must be configurable to a target building with minimal manual effort.

   We believe our simulation system meets all of these listed requirements.

# E    DERIVATION FOR TENSORIZED FINITE DIFFERENCE (FD) EQUATIONS

This appendix describes the method of calculating the flow of heat and the resulting temperatures throughout the building.

## E.1    ASSEMBLING THE ENERGY BALANCE

The fundamental energy balance for a general-purpose closed body is formulated in Equation 6. The first term represents the effects of non-stationary heat dissipation or heat absorption over time over volume of the body. $Q$ represents the energy absorbed or released per unit volume and is a function of the mass and heat capacity of the body. The second term represents thermal flux over the surface of the body, where $\mathbf{n}$ is the unit normal vector of the surface $S$ and $\mathbf{F}$ is the specific energy absorbed or released through the surface. Common modes of thermal flux include conduction, convection, and radiation. The right side of the equation represents the total energy absorbed by the body across the system boundary, or via an external source or sink.

$$\frac{d}{dt} \int_{V(t)} Q dV + \oint_{S(t)} \mathbf{n} \cdot \mathbf{F} dS = \int_{V(t)} P dV \tag{5}$$

To enable computation, we divide the body into small discrete units, called **Control Volumes** (CV), and iteratively calculate temperature on each on each CV using the method of Finite Differences (FD).

We model three modes of heat transfer into each CV: forced convection, conduction, and external source.

Forced convection $Q^{conv}$ is based on energy exchange by moving air (or any other fluid, in general), and conduction, $Q^{cond}$ is the exchange of energy through solid objects, such as walls. External sources (or sinks) $Q^x$ represent the heating or cooling from external devices, such as electric heating coils, diffusers, etc.

Each CV has the capacity to absorb heat over time, which is expressed as $\frac{dU}{dt}$, governed by its heat capacity, $c$.

These factors allow us to construct an energy balance equation that conserves energy $Q^{in} - Q^{out} = \frac{dU}{dt}$.

We assume that the ceilings and floors are adiabatic, fully insulated, not allowing any heat exchange. This reduces the problem to a 2D problem, with 3D control volumes that can only exchange energy laterally.

Our FD objective is to solve for the temperature at each CV within the building, which presents $N$ unknowns and $N$ equations, where $N$ is the number of CVs in the FD grid.

Rather than creating separate spacial cases in the FD equations for exterior, boundary, and interior CVs, we would like to create a single equation that can be computed across the entire grid. This equation can then be tensorized using the Tensorflow matrix library, and accelerated with GPUs or TPUs.

We label each four interacting surfaces of the CV: left = 1, right = 3, bottom = 2, and top = 4.

Then, for a discrete unit of time $\Delta t$ we specify energy exchange across the surfaces as $Q_1, Q_2, Q_3, Q_4$ and adopt the arbitrary, but consistent convention that energy flows into surfaces 1 and 2, and out of surfaces 3, and 4. (Of course, energy can flow the other direction too, but that

will be indicates with a negative value.) Our convention also assumes that external energy flows into the CV.

That allows us to construct the energy balance as:

$$Q^x + Q_1^{cond} + Q_1^{conv} + Q_2^{cond} + Q_2^{conv} - Q_3^{cond} - Q_3^{conv} - Q_4^{cond} - Q_4^{conv} = \frac{dU}{dt} \quad (6)$$

### E.2 COMPUTING HEAT TRANSFER VIA CONDUCTION, CONVECTION, AND THERMAL ABSORPTION

We apply the Fourier's Law of conduction, illustrated in Figure 9, which is the rate of transfer in Watts:

$$\dot{Q}^{cond} = -\frac{kA}{L}\frac{dT}{dt} \quad (7)$$

Which is approximated over the discrete CV as:

$$\dot{Q}^{cond} \approx -\frac{kA}{L}\frac{\Delta T}{\Delta t} \quad (8)$$

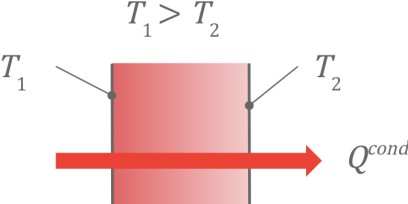

$$T_1 > T_2$$

Figure 9: Conduction Heat Transfer

Where $k$ is the thermal conductivity of the material, $A$ is the flux area perpendicular to the flow of heat, $L$ is the distance traveled through the material, $\Delta T$ is the temperature difference in the source and sink, and $\Delta t$ is a discrete timestep interval.

We can remove the dot (time derivative) by multiplying by discrete unit time, and converting thermal power (energy per unit time) into energy:

$$Q^{cond} \approx -\frac{kA}{L}\frac{\Delta T}{\Delta t} \times 1 = -\frac{kA}{L}\Delta T \quad (9)$$

Let's orient the conductivity equation along the horizontal ($u$) and the vertical directions ($v$).

For the horizontal heat transfer:

$$Q_{1,3}^{cond} = -\frac{kvz}{u}\Delta T(D.5) \quad (10)$$

And for vertical heat transfer:

$$Q_{2,4}^{cond} = -\frac{kuz}{v}\Delta T \quad (11)$$

Where $z$ is the 3rd dimension size, which is the distance from the floor to the ceiling, and $A = vz$ and $A = uz$ for horizontal and vertical flux surface areas.

This is good for modeling heat exchange through solid objects, but we also need to model the heat exchanges from the outside across the boundary to the interior via forced air convection (i.e., wind).

For convection, we'll apply Newton's Law of Cooling, illustrated in Figure 10 for modeling heat transfer via forced air currents across a surface $A$, perpendicular to the flow of heat as:

$$Q^{cond} = -hA\Delta T \tag{12}$$

The negative sign in Equations 7 - 12 are due to the fact that energy flows in the direction opposite of the temperature gradient, $\Delta T$, i.e., from high to low.

Here, $h$ is the convection coefficient and is a function of the amount of air blowing over the exterior surface of the wall.

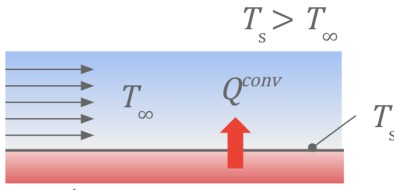

Figure 10: Convection Heat Transfer

We define the three types of CVs:

1. **Exterior CVs** are CVs that represent the outside weather conditions, such as $T_\infty$ , which are note calculated by the FD calculator, just specified by the current input conditions.

2. **Interior CVs** are CVs where all four sides are adjacent to non-exterior CVs (Figure 11).

3. **Boundary CVs** are CVs that share one or two faces with exterior CVs and one two or three faces with interior CVs. These CVs require special handling, since they represent the transfer of energy between the outside and the inside of the building. Boundary CVs that share two sides with the exterior are **Corner CVs** (Figure 12) and boundary CVs that share only one side with an exterior CV are **Edge CVs** (Figure 13).

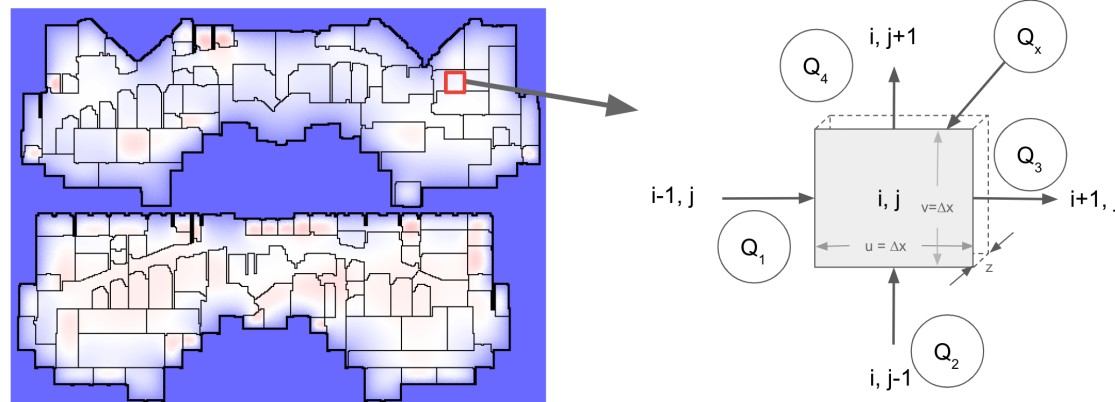

Figure 11: Interior Control Volumes

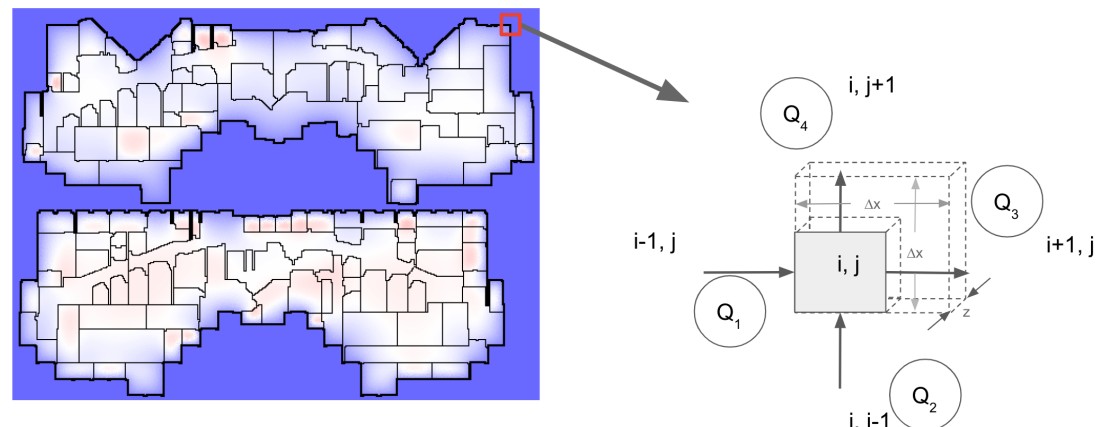

Figure 12: Boundary Corner Control Volumes

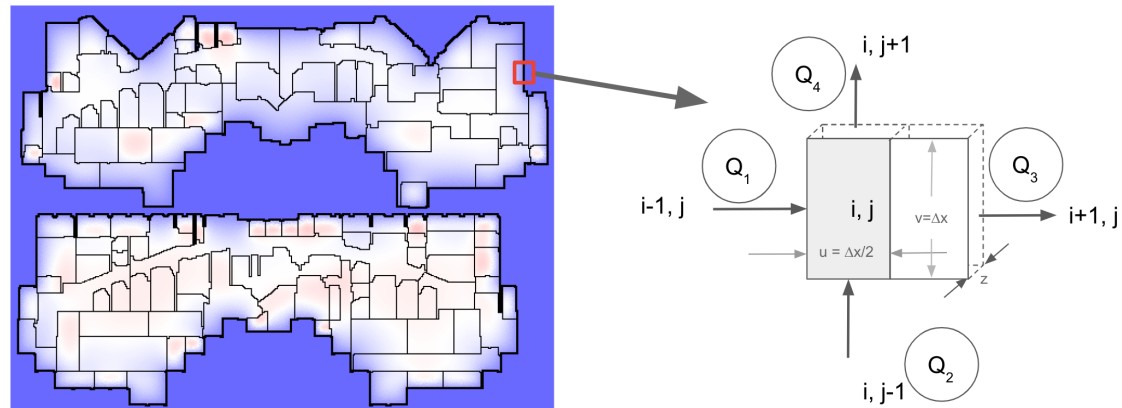

Figure 13: Boundary Edge Control Volumes

The temperatures that are estimated in FD represent the center of the control volume, or its mean. In the case of convection, the temperatures at the exterior surface of the wall us unknown and have to be calculated. Therefore, the center of the Edge CV represents the surface temperature and is split halfway between the outside and inside, where the volume of an edge CV is half of the mass of an interior CV. Similarly, an corner CV is cut in half in both directions, and is one quarter the volume ov an interior CV.

Since we are assuming rectangular CVs, note that $v = v_1 = v_3$, and $u = u_2 = u_4$.

Since outside temperatures and HVAC responses vary, we have a non-stationary thermal system where the flow of energy through the CVs that is not constant. This requires us to evaluate the right-hand term in Equation 6 that allows the volume to absorb or dissipate heat over time, which is governed by the mass $m = \rho V = \rho u v z$, heat capacity $c$ and rate of change of temperature $\frac{dT}{dt}$ .

$$\frac{dU}{dt} = cm\frac{dT}{dt} = c\rho V\frac{dT}{dt} = c\rho u v z\frac{dT}{dt} \tag{13}$$

Equation 13 can be approximated over the small differential CV as:

$$\frac{dU}{dt} \approx c\rho u v z\frac{T_{i,j} - T_{i,j}^{(-)}}{\Delta t} \tag{14}$$

where $T_{i,j}^{(-)}$ is the temperature if the $i, j$ CV at the previous timestep and the timestep interval is $\Delta t$, which can be treated as a fixed parameter.

### E.3 Solving for the temperature at each CV

To enable accelerating the calculation using tensor operations, we would like to define a single equation for all CV that do not require (a) conditionals, (b) for loops, or (c) referencing neighboring CVs. That objective will require the construction of a few auxiliary matrices, and every CV will have convection and conduction components that may be disabled with zero-valued convection and conduction coefficients as appropriate.

Combining the Energy Balance in Equation 5 with the conduction and convection equations (Equations 10-13) we can include all terms for all faces on the $i, j$ CV. Our goal is to solve for $T_{i,j}$ which can then be run over multiple sweeps to convergence.

$$
\begin{aligned}
Q_x - k_1 vz\frac{T_{i,j} - T_{i-1,j}}{u} - h_1 vz(T_{i,j} - T_\infty) - k_2 uz\frac{T_{i,j} - T_{i,j-1}}{v_2} - h_2 vz(T_{i,j} - T_\infty)+ \\
+k_3 vz\frac{T_{i+1,j} - T_{i,j}}{u_3} + h_3 vz(T_\infty - T_{i,j}) + k_4 uz\frac{T_{i,j+1} - T_{i,j}}{v_4} + h_4 vz(T_\infty - T_{i,j}) = \quad (15) \\
= \frac{c\rho uvz}{\Delta t}\left(T_{i,j} - T_{i,j}^{(-)}\right)
\end{aligned}
$$

Next, we want to solve for temperature $T_{i,j}$ by rearranging the terms, which provides a single equation that can be used to calculate CV temperatures for both boundary and interior CVs.

$$
T_{i,j} = \frac{Q_x + vz\left[\frac{k_1}{u}T_{i-1,j} + h_1 T_\infty + \frac{k_3}{u}T_{i+1,j} + h_3 T_\infty\right] + uz\left[\frac{k_2}{v}T_{i,j-1} + h_2 T_\infty + \frac{k_4}{v}T_{i,j+1} + h_4 T_\infty\right] + \frac{c\rho uvz}{\Delta t}T_{i,j}^{(-)}}{vz\left[\frac{k_1}{u} + h_1 + \frac{k_3}{u} + h_3\right] + uz\left[\frac{k_2}{v} + h_2 + \frac{k_4}{v} + h_4\right] + \frac{c\rho uvz}{\Delta t}}
$$
$$(16)$$

### E.4 Tensorizing the temperature estimate

Equation 16 can be used iterative, but to exploit the acceleration from matrix operations on GPUs and TPUs using the TensorFlow Library, we'll want to reshape the equation slightly for a single tensor pipeline that doesn't iterate over individual CVs.

Furthermore, we can avoid referencing neighboring temperatures $(T_{i-1,j}, T_{i+1,j}, T_{i,j-1}, T_{i,j+1})$ in the pipeline by creating four *shifted* temperature Tensors, $T_1 = \text{shift}(T, 3)$, $T_3 = \text{shift}(T, \text{LEFT})$, $T_2 = \text{shift}(T, \text{UP})$, $T_4 = \text{shift}(T, \text{DOWN})$.

We can also frame oriented conductivity as a Tensors left $K_1$, right $K_3$, below $K_2$, above $K_4$, where:

$$
k_{1,i,j} = \begin{cases} k_{i,j} & \text{CVs at } i, j \text{ and } i-1, j \text{ are interior or boundary} \\ 0 & \text{otherwise} \end{cases} \quad (17)
$$

$$
k_{3,i,j} = \begin{cases} k_{i,j} & \text{CVs at } i, j \text{ and } i+1, j \text{ are interior or boundary} \\ 0 & \text{otherwise} \end{cases} \quad (18)
$$

$$
k_{2,i,j} = \begin{cases} k_{i,j} & \text{CVs at } i, j \text{ and } i, j-1 \text{ are interior or boundary} \\ 0 & \text{otherwise} \end{cases} \quad (19)
$$

$$
k_{4,i,j} = \begin{cases} k_{i,j} & \text{CVs at } i, j \text{ and } i, j+1 \text{ are interior or boundary} \\ 0 & \text{otherwise} \end{cases} \quad (20)
$$

Note that the conductivity matrix $K$ is a fixed input parameter for the building.

Applying the same reasoning, we can generate four oriented convection Tensors, $H_1, H_2, H_3, H_4$ as:

$$h_{1,i,j} = \begin{cases} h & \text{CV at } i,j \text{ is boundary and CV at } i-1,j \text{ is exterior} \\ 0 & \text{otherwise} \end{cases} \tag{21}$$

$$h_{3,i,j} = \begin{cases} h & \text{CV at } i,j \text{ is boundary and CV at } i+1,j \text{ is exterior} \\ 0 & \text{otherwise} \end{cases} \tag{22}$$

$$h_{2,i,j} = \begin{cases} h & \text{CV at } i,j \text{ is boundary and CV at } i,j+1 \text{ is exterior} \\ 0 & \text{otherwise} \end{cases} \tag{23}$$

$$h_{4,i,j} = \begin{cases} h & \text{CV at } i,j \text{ is boundary and CV at } i,j-1 \text{ is exterior} \\ 0 & \text{otherwise} \end{cases} \tag{24}$$

Note that $h$ is a time-dependent constant that represents the amount of airflow over the surface of the building, assumed to be uniformly applied on all exterior walls of the building.

Finally, we classify each boundary CV as `TOP-LEFT CORNER`, `TOP-RIGHT CORNER`, `BOTTOM-LEFT CORNER`, `BOTTOM-RIGHT CORNER` or `LEFT EDGE`, `RIGHT EDGE`, `TOP EDGE`, or `BOTTOM EDGE` in order to form Tensors $U$ and $V$, which are the CV widths and heights.

$$u_{i,j} = \begin{cases} \frac{\Delta x}{2} & \text{CV at } i,j \text{ is BOUNDARY and ANY CORNER or TOP or BOTTOM EDGE} \\ \Delta x & \text{otherwise} \end{cases} \tag{25}$$

$$v_{i,j} = \begin{cases} \frac{\Delta x}{2} & \text{CV at } i,j \text{ is BOUNDARY and ANY CORNER or LEFT or RIGHT EDGE} \\ \Delta x & \text{otherwise} \end{cases} \tag{26}$$

where $\Delta x$ is the fixed horizontal and vertical dimension of an `INTERIOR` CV.

Now we can complete the Tensor expression of the FD equation:

$$\mathbf{T} = \left[ Q_x + Vz \left[ K_1 U^{-1} T_1 + H_1 T_\infty + K_3 U^{-1} T_3 + H_3 T_\infty \right] + Uz \left[ K_2 V^{-1} T_2 + H_2 T_\infty + K_4 V^{-1} T_4 + H_4 T_\infty \right] + \frac{CPUVz}{\Delta t} T^{(} \\ \left[ Vz \left[ K_1 U^{-1} + H_1 + K_3 U^{-1} + H_3 \right] + Uz \left[ K_2 V^{-1} + H_2 + K_4 V^{-1} + H_4 \right] + \frac{CPUVz}{\Delta t} \right]^{-1}$$

For each timestep, we execute Equation E.4 as single-step tensor operations until convergence, where the maximum change across all CVs between current and last iteration is less then a conservative lower threshold, $\epsilon \leq 0.01^\circ C$

## F    SIMULATOR CONFIGURATION PROCEDURE DETAILS

To configure the simulator, we require two type of information on the building:

1. Floorplan blueprints. This includes the size and shapes of rooms and walls for each floor.

2. HVAC metadata. This includes each device, its name, location, setpoints, fixed parameters, and purpose.

We preprocess the detailed floorplan blueprints of the building and extract a grid that gives us an approximate placement of walls and how rooms are divided. This is done via the following procedure:

1. Using threshold $t$, binarize the floorplan image into a grid of 0s and 1s.

2. Find and replace any large features that need to be removed (such as doors, a compass, etc)

3. Iteratively apply standard binary morphology operations (erosion and dilation) to the image to remove noise from background, while preserving the walls.

4. Resize the image, such that each pixel represents exactly one control volume

5. Run a connected components search to determine which control volumes are exterior to the building, and mark them accordingly

6. Run a DFS over the grid, and reduce every wall we encounter to be only a single control volume thick in the case of interior wall, and double for exterior wall

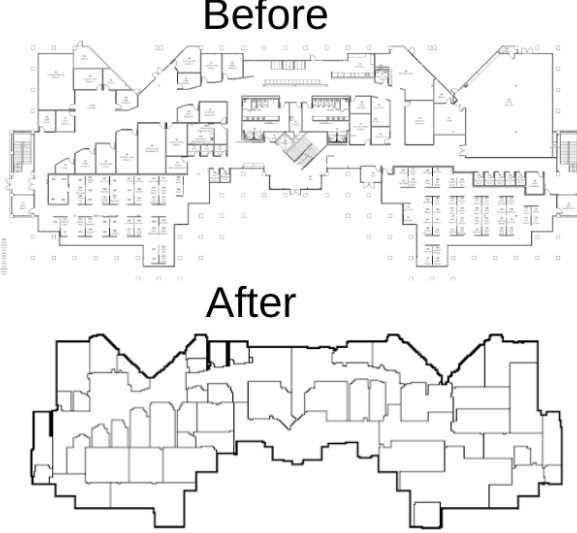

Figure 14: Before and after images of the floorplan preprocessing algorithm

We also employ a simple user interface to label the location of each HVAC device on the floorplan grid. This information is passed into our simulator, and a custom simulator for the new building, with roughly accurate HVAC and floor layout information, is created. This allows us to then calibrate this simulator using the real world data, which will now match the simulator in terms of device names and locations.

We tested this pipeline on SB1, which consisted of two floors with combined surface area of 93,858 square feet, and has 173 HVAC devices. Given floorplans and HVAC layout information, a single technician was able to generate a fully specified simulation in under three hours. This customized simulator matched the real building in every device, room, and structure.

# G    CALIBRATION HYPERPARAMETER TUNING DETAILS

The hyperparameter tuning was performed over a seven day period on 200 CPUs.

Table 3: Thermal properties that were set by the calibration process, with min/max bounds and selected values.

| HYPERPARAMETER | MIN | MAX | BEST |
|---|---|---|---|
| CONVECTION_COEFFICIENT $(W/m^2/K)$ | 5 | 800 | 357 |
| EXTERIOR_CV_CONDUCTIVITY $(W/m/K)$ | 0.01 | 1 | 0.83 |
| EXTERIOR_CV_DENSITY $(kg/m^3)$ | 0 | 3000 | 2359 |
| EXTERIOR_CV_HEAT_CAPACITY $(J/Kg/K)$ | 100 | 2500 | 2499 |
| INTERIOR_WALL_CV_CONDUCTIVITY $(W/m/K)$ | 5 | 800 | 5 |
| INTERIOR_WALL_CV_DENSITY $(kg/m^3)$ | 0.5 | 1500 | 1500 |
| INTERIOR_WALL_CV_HEAT_CAPACITY $(J/Kg/K)$ | 500 | 1500 | 1499 |
| SWAP_PROB | 0 | 1 | 0.003 |
| SWAP_RADIUS | 0 | 50 | 50 |

# H    ADDITIONAL SPATIAL ERROR VISUALIZATIONS

Here we present some other visuals that may be enlightening.

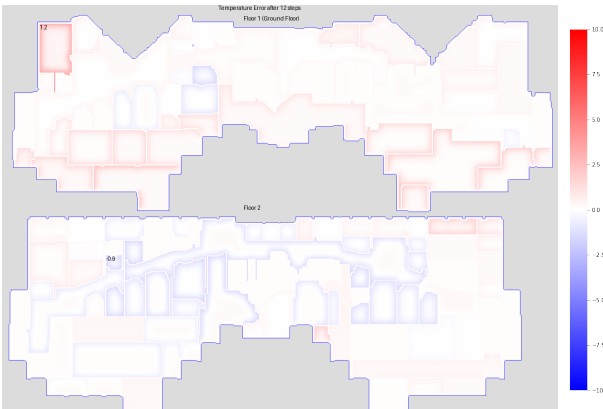

Figure 15: Visualization of simulator drift after only a single hour, on the validation data. As can be clearly seen, at this point there is almost no error.

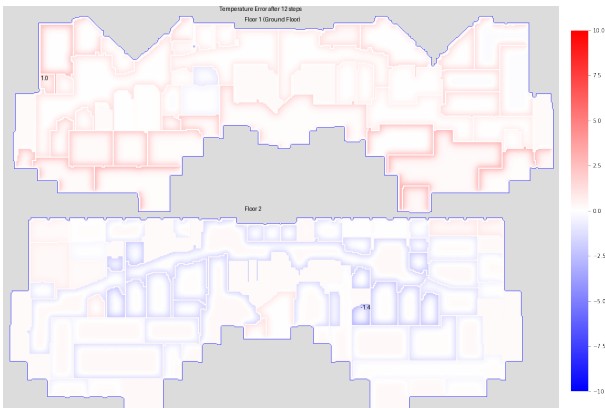

Figure 16: Visualization of simulator drift after only a single hour, on the train data. Again, there is almost no error.

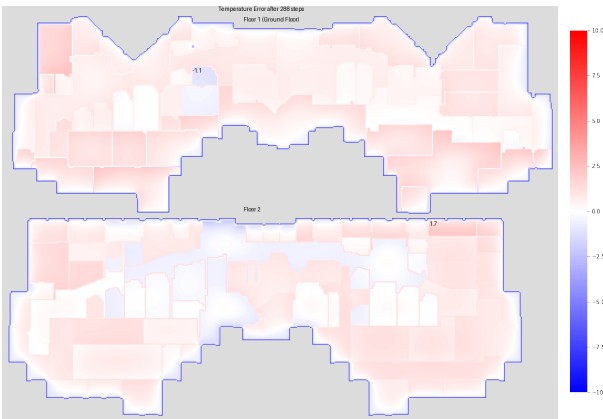

Figure 17: Visualization of simulator drift after one day, on the train data.

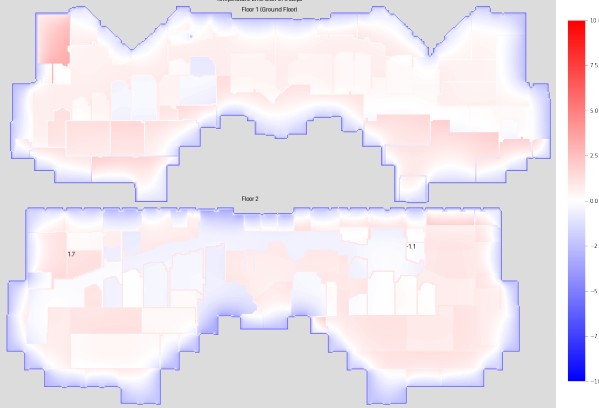

Figure 18: Visualization of simulator drift after two days, on the train data. Interestingly, this looks better than it did after only one day.

# I  ADDITIONAL MODNN DETAILS

## I.1  DETAILED MODNN MODEL STRUCTURE

In the ModNN model, a Gated Recurrent Unit (GRU) module is used to capture the highly nonlinear relationships between disturbance variable inputs and heat gains, while a Fully Connected (FC) neural network module captures the HVAC input. Notably, the HVAC input in this study is air-side, allowing it to be directly added to the disturbance variables. However, this module can be extended to accommodate different HVAC systems. For instance, a radiation-based HVAC system would require a Recurrent Neural Network (RNN) module to consider the heat lagging effects.

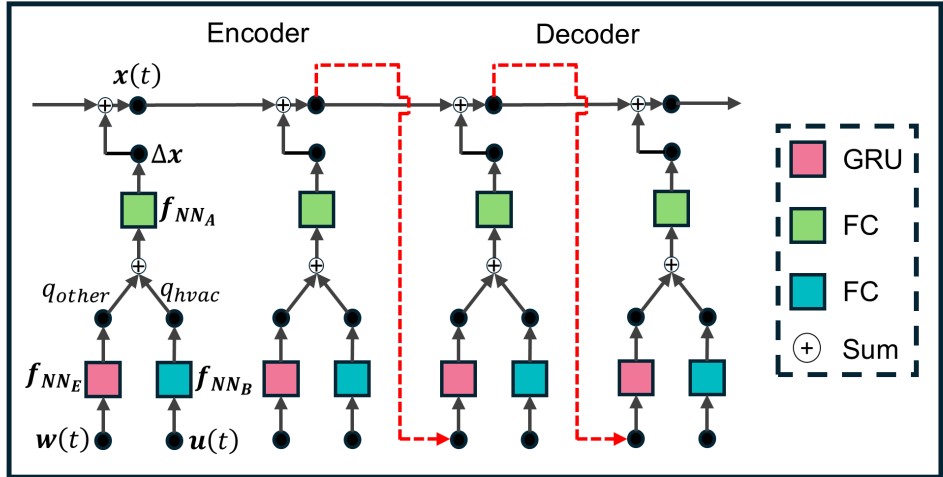

Figure 19: Diagram of Detailed ModNN Model Structure.

The integrated heat transfer terms will go through another FC module to model the dynamics of zone air mass and heat capacity. The output represents the temperature change per timestep, which is recursively added to the previous timestep's temperature for future predictions.

## I.2  MODEL PERFORMANCE

As shown in Figure 20, the gray line represents the predicted space air temperature, which closely matches the measured data, achieving an MAE of 0.3 °C for 24-hour predictions. To demonstrate that the proposed model can not only predict accurately but also capture the impact of changes in control input (HVAC power), we conducted a sanity check. In this test, the colored lines, from blue to red, represents the intentionally adjusted HVAC inputs from 4 kW to -4 kW. The response of the PI-ModNN model adhered to physical principles: additional heating resulted in an increase in space air temperature, while additional cooling led to a decrease. This confirms that the model reliably responds to variations in control input, which is essential for enabling a reinforcement learning (RL) agent to explore and optimize control actions.

We also compared the MMD of the LSTM and ModNN models, as illustrated in Figure 21. The X-axis represents one-step HVAC load changes (negative values indicate increased cooling, positive values indicate reduced cooling), and the Y-axis shows the corresponding changes in space air temperature. Each scatter point reflects data under varying weather and occupancy conditions.

From the black points (ground truth), we observe a clear decreasing trend in space air temperature as the cooling load increases. The ModNN model, represented by blue points, captures a similar trend. However, the LSTM model, depicted by red points, shows an incorrect trend where the space air temperature increases with additional cooling. This discrepancy is further highlighted by the difference in distributions between the ground truth (black contour plot) and the LSTM model (red contour plot). The MMD of the LSTM model to the ground truth is 0.14, which is significantly higher than the MMD of the ModNN model to the ground truth (0.05).

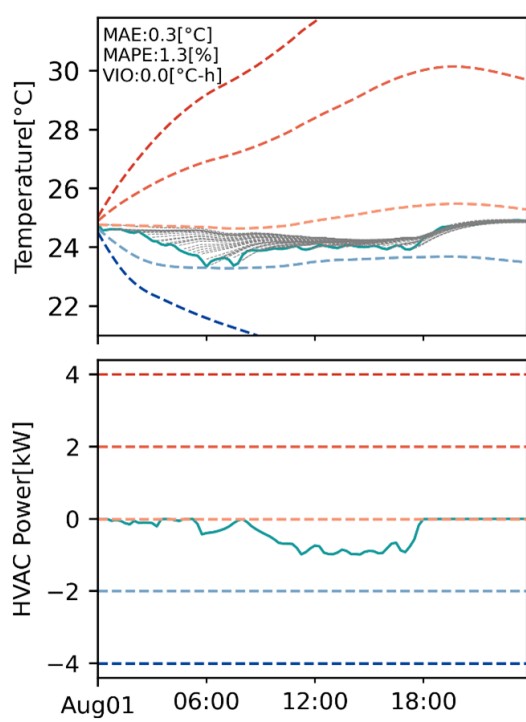

Figure 20: PI-ModNN model performance.

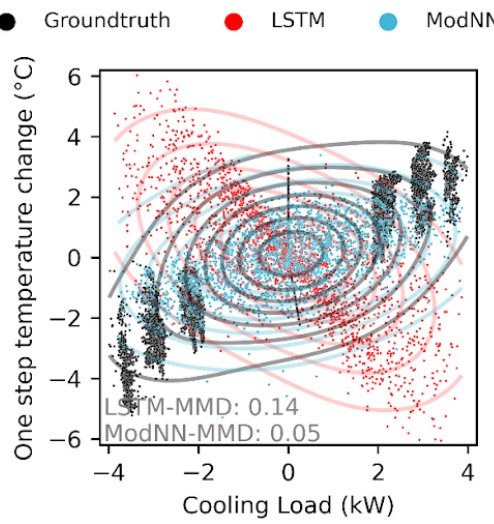

Figure 21: Maximum mean discrepancy of LSTM and ModNN.

### I.3 CONTROL OPTIMIZATION BASED ON PI-MODNN

Based on the proposed model, we applied optimal control to evaluate the potential for energy savings. The green line represents the baseline scenario, where unnecessary cooling occurred in the early morning. In contrast, the red line shows the results of the optimal control strategy. As shown, the HVAC system remains off during the early morning hours to allow the space air temperature free floating, and the space temperature consistently stays close to the upper comfort limit. This strategy achieves significant energy savings, reducing air-side energy consumption by over 64%.

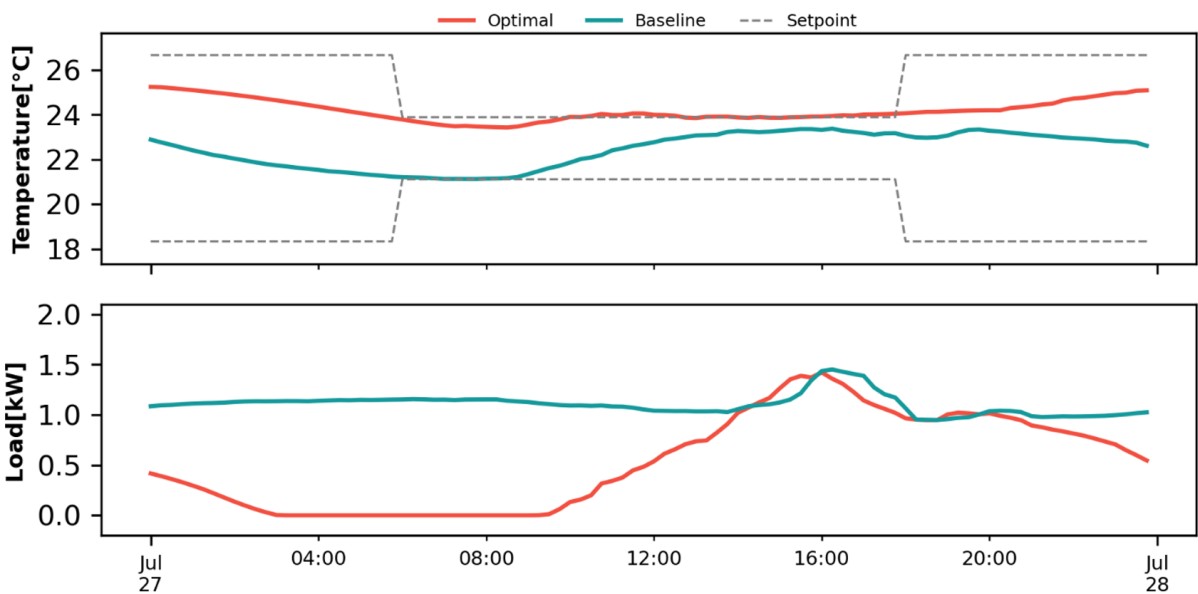

Figure 22: Optimal Control for an Energy Saving Case Study.

We then applied the same approach to a flexibility case study, aiming to shift energy consumption from peak hours to off-peak hours while maintaining thermal comfort. The overall results for one month are shown below. In the figure, the red box indicates the baseline case, and the blue box represents the optimal control results, with the gray shaded area marking the peak hours. The space temperature was generally well maintained, and peak demand was reduced through pre-cooling. However, some temperature violations were observed due to the shifted load. This suggests that future work could incorporate hard constraints or fine-tune the balance parameters to better manage the trade-off between load shifting and comfort.

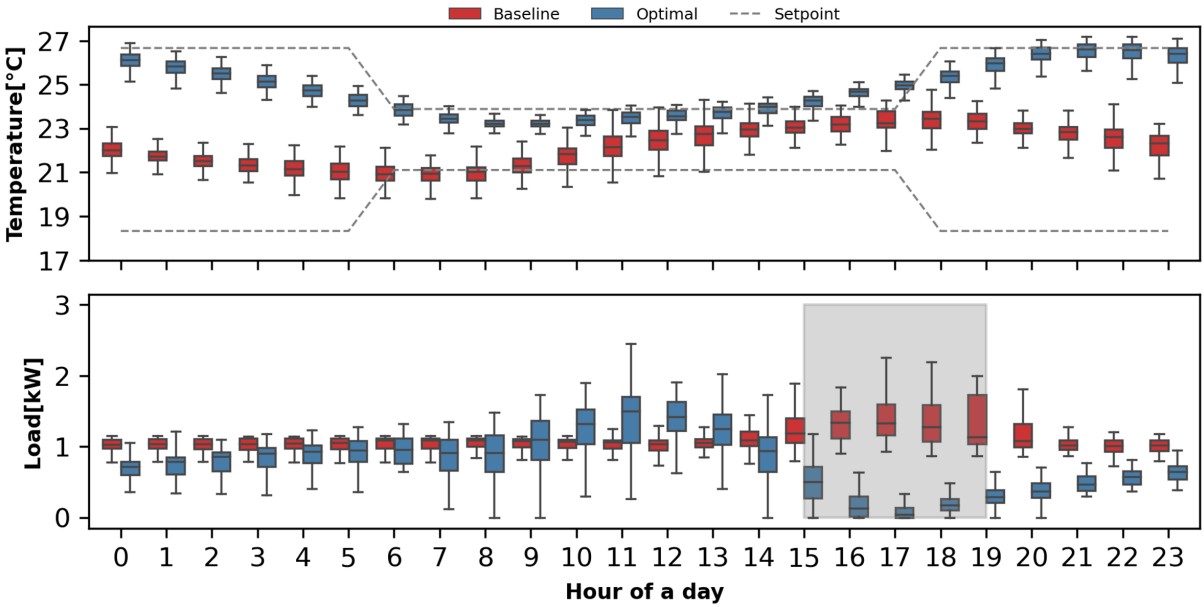

Figure 23: Optimal Control for an Load Shifting Case Study.

### I.4   ADJACENT MATRIX FOR MULTIZONE DYNAMIC MODELING

To extend the single-zone model to a multi-zone framework, we developed a conduction heat transfer model to capture interactions between adjacent zones. A skew-symmetric matrix is used to represent temperature differences between adjacent zones, as shown in the figure. Based on the heat conduction equation:

$$Q = \frac{kA\Delta(T_i - T_j)}{l} := f_{NN_{adj}}(T_i - T_j)$$

Where $q$ represents heat transfer through conduction, $k$ is the thermal conductivity of the material, $A$ is the cross-sectional area, $T_i$ and $T_j$ are the space air temperatures of the adjacent zones, and $l$ is the thickness of the wall.

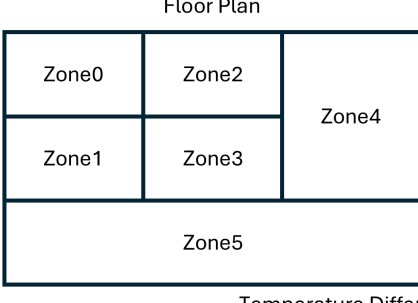

**Floor Plan**

| | | |
|---|---|---|
| Zone0 | Zone2 | |
| Zone1 | Zone3 | Zone4 |
| Zone5 | | |

**Adjacent Matrix**

```
0 1 1 0 0 0
1 0 0 1 0 1
1 0 0 1 1 0
0 1 1 0 1 1
0 0 1 1 0 1
0 1 0 1 1 0
```

**Temperature Difference**

| | | | | | |
|---|---|---|---|---|---|
| $0$ | $T_1 - T_0$ | $T_2 - T_0$ | $0$ | $0$ | $0$ |
| $T_0 - T_1$ | $0$ | $0$ | $T_3 - T_1$ | $0$ | $T_5 - T_1$ |
| $T_0 - T_2$ | $0$ | $0$ | $T_3 - T_2$ | $T_4 - T_2$ | $0$ |
| $0$ | $T_1 - T_3$ | $T_2 - T_3$ | $0$ | $T_4 - T_3$ | $T_5 - T_3$ |
| $0$ | $0$ | $0$ | $T_3 - T_4$ | $0$ | $T_5 - T_4$ |
| $0$ | $T_1 - T_5$ | $0$ | $T_3 - T_5$ | $T_4 - T_5$ | $0$ |

Figure 24: Adjacency Matrix for Extending from Single-Zone to Multi-Zone.

This heat transfer term is learned using another FC layer and then integrated with the other heat transfer terms to predict the next step's space air temperature.

### I.5   DATA STRUCTURE

The input dimension of the encoder is

$$(D_{\text{state}} + D_{\text{dis}} + D_{\text{adj}} + D_{\text{hvac}}) \times B \times L_{\text{en}},$$

where $D_{\text{state}}$, $D_{\text{dis}}$, $D_{\text{adj}}$, and $D_{\text{hvac}}$ are the feature dimensions of the state variables (the number of zones), disturbance variable, number of adjacent zones, and control variables, respectively. $B$ is the batch size, and $L_{\text{en}}$ is the length of the encoder.

The input dimension of the decoder is

$$(D_{\text{dis}} + D_{\text{adj}} + D_{\text{hvac}}) \times B \times L_{\text{de}},$$

since the future indoor air temperature is unknown at timestep $t$. Thus, the input dimension for the decoder excludes $D_{\text{state}}$.

The output dimension is

$$1 \times B \times L_{\text{de}}.$$

The input vector can be formulated as shown below:

$$
\begin{bmatrix}
w_{t-L_{en}}^{Solar} & w_{t-L_{en}+1}^{Solar} & \cdots & w_t^{Solar} & \cdots & w_{t+L_{de}-2}^{Solar} & w_{t+L_{de}-1}^{Solar} & w_{t-L_{en}}^{T_{amb}} & w_{t-L_{en}+1}^{T_{amb}} & \cdots & w_t^{T_{amb}} & \cdots \\[2mm]
w_{t+L_{de}-2}^{T_{amb}} & w_{t+L_{de}-1}^{T_{amb}} & w_{t-L_{en}}^{Time} & w_{t-L_{en}+1}^{Time} & \cdots & w_t^{Time} & \cdots & w_{t+L_{de}-2}^{Time} & w_{t+L_{de}-1}^{Time} & w_{t-L_{en}}^{Occ} & w_{t-L_{en}+1}^{Occ} & \cdots & w_t^{Occ} \\[2mm]
\cdots & w_{t+L_{de}-2}^{Occ} & w_{t+L_{de}-1}^{Occ} & u_{t-L_{en}}^{HVAC} & u_{t-L_{en}+1}^{HVAC} & \cdots & u_t^{HVAC} & \cdots & u_{t+L_{de}-2}^{HVAC} & u_{t+L_{de}-1}^{HVAC} & x_{t-L_{en}}^{Zone} & x_{t-L_{en}+1}^{Zone} & \cdots & x_t^{Zone} & \cdots
\end{bmatrix}
$$

Here, we define $w_t^k \in \mathbb{R}^{\text{Disturbance} \times \text{batch} \times (L_{\text{en}}+L_{\text{de}}) \times D}$ as the disturbance vector, where $t$ is the timestep and $k$ is the feature category. The control input term is represented as

$$u_t^k \in \mathbb{R}^{1 \times \text{batch}(L_{\text{en}}+L_{\text{de}}) \times 1}$$

and the state variable, representing space air temperature in building dynamics modeling, is given by

$$x_t^k \in \mathbb{R}^{1 \times \text{batch} \times (L_{\text{en}}+1) \times 1}.$$

We further clarify the abbreviations for $k$ used in this matrix:
- $solar$: global solar radiation
- $time$: time information
- $T_{\text{OA}}$: outside air temperature
- $Occ$: occupant number

## I.6 HYPERPARAMETERS

We summarized all the hyperparameters used in this study as shown below. The input vector can be formulated as shown below:

| Parameters | Value Setting |
|---|---|
| Encoder Length | 16,48,96 |
| Decoder Length (Prediction Horizon) | 2,4,8,16,24,48,96 |
| Batch Size | 128,256,512,1024 |
| Encoder Hidden Size | 16,20,24 |
| Decoder Hidden Size | 16,20,24 |
| Training Epochs | 300,400,500 |
| Learning Rate | 0.001,0.0001 |
| Activation Function | $\sigma$, tanh, Relu |
| Optimizer | Adam |
| Disturbance Module Input Dimension | 6 |
| Disturbance Module Hidden Dimension | 10,15,20 |
| Disturbance Module Output Dimension | 1 |
| HVAC Module Input Dimension | 1 |
| HVAC Module Hidden Dimension | 5,10,15 |
| HVAC Module Output Dimension | 1 |
| State Module Dimension | 1 |
| State Module Dimension | 5,10,15 |
| State Module Dimension | 1 |

## J    SIMULATOR SAC AGENT TRAINING DETAILS AND PERFORMANCE ANALYSIS

We will now go into more details on the simulator RL agent training and performance as compared to the baseline.

Each agent was trained on a single CPU. We restricted the action space to supply air and water temperature setpoints. For the observation space, we found that providing the agent with the dozens of temperature sensors was too much noisy information and not useful. Instead, we provided the agent with a histogram, grouping temperatures into 1° Celsius bins, ranging from 12° to 30°, and calculating the frequency of each bin. The tallies are then normalized and provided as part of the observation. This led to much better performance.

Our reward function is a weighted, linear combination of the normalized carbon footprint, cost, and comfort levels within the building. While an 8% improvement over the baseline on this scalar reward is significant, we can see the improvements of the SAC agent over the baseline even more clearly when we break down these factors further into physical measures.

For this analysis, we break down the reward into four components that contribute to it, and see how the learned policy compares with the baseline. The components are: setpoint deviation, carbon emissions, electrical energy, and natural gas energy.

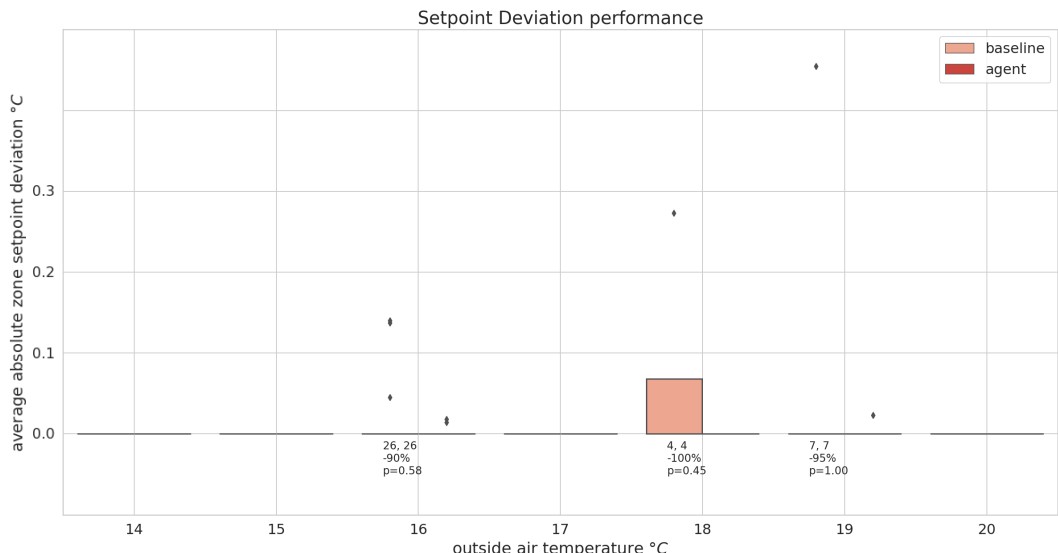

Figure 25: Setpoint Deviation Performance as a function of outside air temperature, which evaluates how well the agent meets comfort conditions compared to the baseline. It is measured as the average number of $°C$ above or below setpoint for all zones in the building. For each outside air degree increment, we include the number of observations for baseline and agent, the percentage change as (baseline - agent) / baseline, and its associated p-score.

Above we display how the baseline and agent compare when it comes to setpoint deviation, the comfort component of the reward function. We show the distribution of deviations grouped by outside air temperatures. While both policies have very minimal setpoint deviation to begin with, the agent strictly improves over the baseline here.

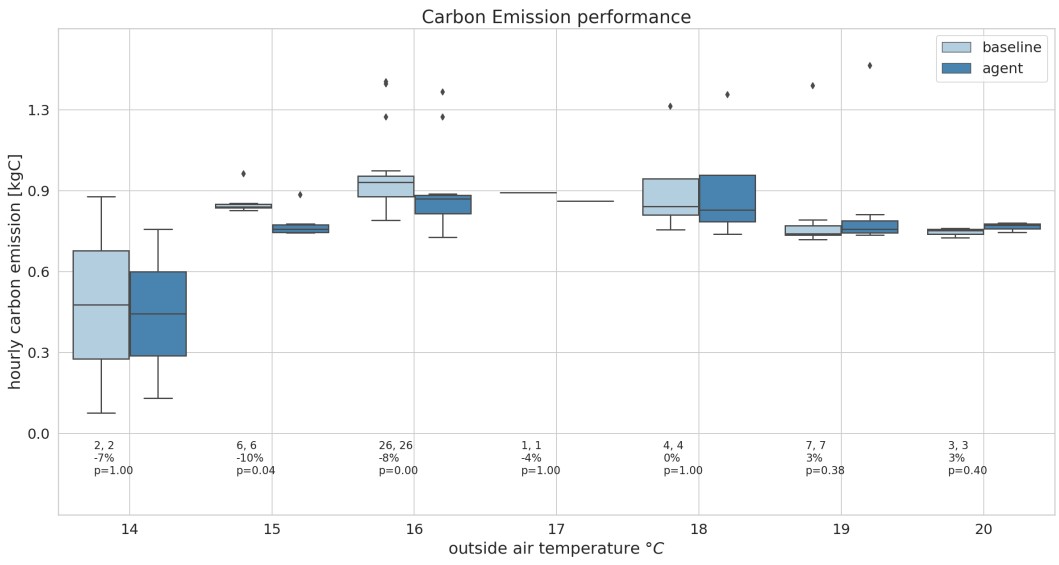

Figure 26: Carbon Emission measures how the agent performs compared to the baseline in terms of the amount of greenhouse gas released from consuming natural gas and electricity. C is combined mass (kgC, or kg Carbon) emitted by non-renewable electricity and natural gas. For each outside air degree increment, we include the number of observations for baseline and agent, the percentage change as (baseline - agent) / baseline, and its associated p-score.

The carbon performance of the agent, as compared with the baseline, is impressive as well. In the temperature range $14°C$ to $18 °C$, the agent is strictly better, and while it is slightly worse for the warmer temperatures, clearly it is a net improvement over the baseline.

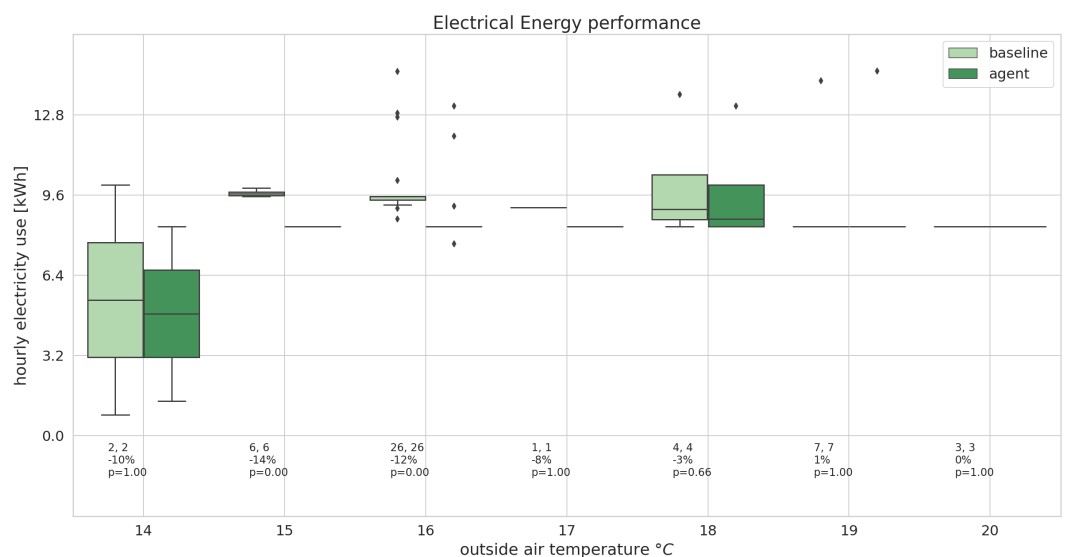

Figure 27: Electrical Energy Performance measured in energy units (kWh) over a fixed interval for both the agent and the baseline policies. For each outside air degree increment, we include the number of observations for baseline and agent, the percentage change as (baseline - agent) / baseline, and its associated p-score.

Once again, when it comes to electric performance, the SAC agent is almost strictly better under all temperature ranges.

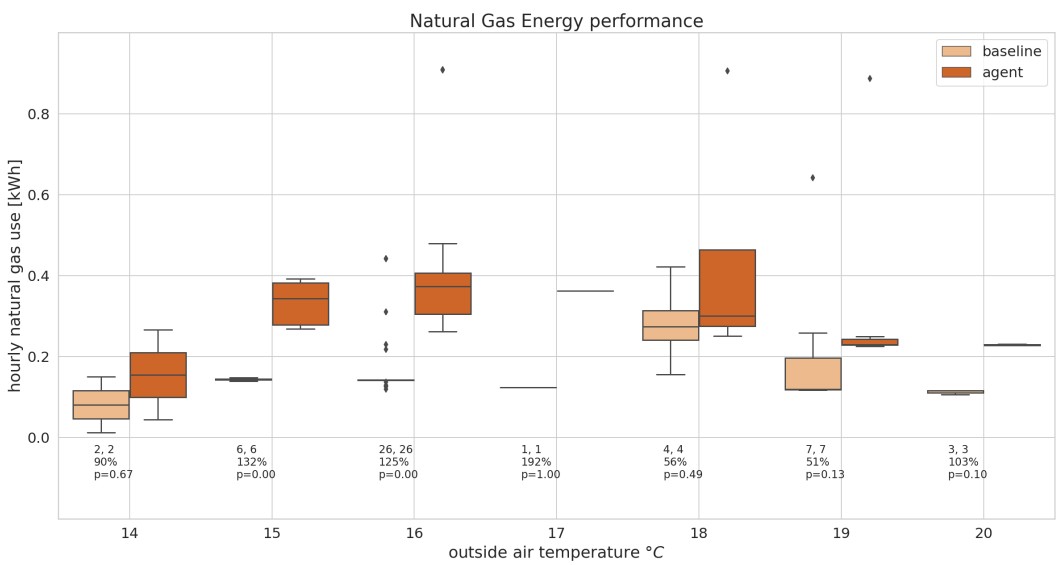

Figure 28: Natural Gas Performance measured in energy units (therm) over a fixed interval for both the agent and the baseline policies. For each outside air degree increment, we include the number of observations for baseline and agent, the percentage change as (baseline - agent) / baseline, and its associated p-score.

Interestingly, the agent converged on a policy that reduced overall carbon emission while increasing natural gas consumption. This is due to the fact that electricity is generated from non-renewable sources and per unit energy, is significantly more expensive than gas.

