# OpenReview forum: "The Smart Buildings Control Suite: A Diverse Open Source Benchmark to Evaluate and Scale HVAC Control Policies for Sustainability"
_ICLR.cc/2026/Conference — ICLR 2026 Conference Withdrawn Submission_

### Official Review · Reviewer_8ufe · 2025-10-31

**Soundness:** 3
**Presentation:** 3
**Contribution:** 3
**Rating:** 4
**Confidence:** 4

**Summary:**

This paper describes "The Smart Buildings Control Suite" (SBCS): a benchmark intended to evaluate and scale HVAC control policies with an explicit focus on generalization across buildings. The suite comprises three parts: (1) a real‑world dataset from 11 commercial buildings (three Building Management Systems BMS vendors) spanning six years; (2) a lightweight, data‑calibrated simulator based on 2‑D finite‑difference heat diffusion tied to HVAC components; and (3) a physically informed neural network (PINN) / ModNN building model designed for control. The benchmark is Gym‑compatible, and the dataset is available via TFDS, with code anonymously supplied. The authors also propose a 3C reward combining comfort, cost, and carbon.

**Strengths:**

- The papers claim to novelty is a three‑part benchmark that combines public historical data, a calibratable interactive simulator, and a control‑oriented PINN is, to my knowledge, novel in this domain. It aims to unlock interactive policy learning while preserving ties to real buildings—something most prior datasets/simulators don’t offer at once. (Sec. 2 contrasts with EnergyPlus/BEAR and single‑building datasets; Sec. 4–6 describe the three pillars.)

- The simulator’s calibration/evaluation protocol is clearly defined and illustrated with quantitative drift. The discussion of parameter plausibility and the compensatory role of the convection coefficient is transparent.

 - The PINN model is satisfactory - It incorporates physical priors and positivity constraints on control gains, and it is validated both by MAE and behavioral sanity (response monotonicity; MMD vs. LSTM).

- The 3C reward seems thoughtfully constructed and extensible (comfort, cost, carbon; with an optional IAQ term and feasibility constraints discussed on 18–20). The comfort loss is tied to literature and occupancy modeling.

 - The paper is well written and includes useful diagrams and visual diagnostics The appendices provide substantial implementation detail on FD derivations, tensorization, preprocessing floor plans and hyperparameters, aiding reproducibility.

- Climate/energy impacts are substantial, and reproducible benchmarks are currently a bottleneck for ML+HVAC. A credible public suite that the community can train on, interactively, could catalyze progress analogous to what Atari/GLUE did in their areas.

**Weaknesses:**

The main concern is that there is *limited evidence* for cross‑building generalization (the central motivation). Despite the stated aim (“solutions that generalize across buildings”), empirical evaluations are are a little different:

- Simulator calibration is primarily shown for SB1 the ModNN is trained/evaluated on SB4

- RL improvements are presented on the simulator, apparently for a single building (Appendix J).

- There is no systematic “train‑on‑subset, test‑on‑held‑out‑buildings” study or adaptation study (fine‑tune few‑shots, etc.).

This is a major gap relative to the benchmark’s thesis.

**Questions:**

The main premise of the paper (generalizability to new buildings) is not validated. Therefore, can you add experiments that train on some buildings and test on unseen buildings, both for the simulator‑trained RL policies and for the PINN? This will substantially address a critical drawback in the results.

---

### Official Review · Reviewer_XvWV · 2025-11-01

**Soundness:** 2
**Presentation:** 3
**Contribution:** 2
**Rating:** 2
**Confidence:** 4

**Summary:**

This paper proposes a benchmark for energy optimization solutions for HVAC systems in large buildings instrumented with various sensors. The authors provide a real-world dataset (11 buildings, 6 years of data, across 3 different environments), a physical simulator to generate additional data from specific buildings, and an alternative simulator based on a Physics-Informed Neural Network (PINN) model.
The paper is clearly written, and the ideas are easy to follow.

**Strengths:**

S1. The problem of optimizing energy consumption in buildings is clearly formulated – using a Markov Decision Process

S2. The physical simulator (and the PINN-based alternative) are valuable tools for generating data for the benchmark. The authors have done a solid job in designing a scalable solution capable of supporting a large number of buildings, which is essential for the benchmark’s robustness and representativeness.

**Weaknesses:**

W1. (the most important weakness point) The work does not appear to be fully finalized. A practical demonstration of the benchmark’s use in evaluating or comparing optimization approaches is expected to assess its actual usefulness. The authors mention at the end of the paper: “We do not have results on training a model and deploying it on these buildings, something we leave for future work.” --> this missing part is essential to validate the proposed benchmark and demonstrate its applicability.

W2. It is not clear why the PINN module is proposed in this paper. The physical simulator already proposed appears sufficient for the purpose of the benchmark. The added value or necessity of introducing a second simulator (PINN) should be clarified.

W3. The dataset is not made available for the review process. While the authors provide the Python code, the six years of real-world data are missing, which prevents reviewers from verifying or reproducing the results.

**Questions:**

The paper presents a promising contribution; however, the absence of experimental validation through concrete use cases (W1) prevents assessing the actual usefulness of the proposed benchmark.

---

### Official Review · Reviewer_vtTJ · 2025-11-01

**Soundness:** 2
**Presentation:** 3
**Contribution:** 2
**Rating:** 2
**Confidence:** 3

**Summary:**

This paper proposes a smart building control suite, open source real world data and physical based simulators and PINN simulators. The control suite has been integrated to GYM for RL development and testing.

**Strengths:**

1.the paper is easy to follow and well structured.
2. The integration to Gym environment is useful to the RL HVAC community
3. The experiment/numbers/figures look solid

**Weaknesses:**

1. The authors claim this tool is easy to scale and transfer across buildings. However it still requires manual process which would become bottleneck in the development.
2. The simulation error is evaluated by the MAE MMD etc, these metrics are internal to the building dynamics and simulators. However it is unclear about the modeling error with controllers.
3. The control suite itself is relatively small with only 11 building, only in NA
4. Looks like the authors haven’t thought about heat transfer and airflow dynamics
5. The authors didn’t compare their simulators to the existing ones such as EnergyPlus.

**Questions:**

See above weakness

---

### Official Review · Reviewer_9fvK · 2025-11-05

**Soundness:** 1
**Presentation:** 1
**Contribution:** 1
**Rating:** 2
**Confidence:** 4

**Summary:**

This paper introduces The Smart Buildings Control Suite, an open-source benchmark for evaluating HVAC control algorithms at scale. It combines real-world HVAC datasets from 11 buildings across North America, and a lightweight, data-calibrated simulator based on finite-difference thermal modeling, together with a physics-informed neural network (PINN) model for building dynamics. The benchmark is designed to facilitate reinforcement learning (RL) research in sustainable building control, promoting generalizable solutions across diverse building types.

**Strengths:**

The paper contains real-world building environments, and provides a detailed illustration of the environment.

**Weaknesses:**

The novelty of this work is rather limited. In the literature, there have been many building datasets, and novel RL algorithms applied to building HVAC. The paper provides a case study, rather than providing any methodological or simulation insights.

The training results on limited RL agents are only briefly summarized without quantitative performance metrics.

Section 7 is unclear about the settings and motivations.

Equation (2), is the variable a scalar or vector? What is the difference compared to RC dynamics or other ODE-based simulation? It is quite unclear.

What is the unit of Fig 1's axis? Fig 3's captions are very small and could not real.

**Questions:**

Please see weaknesses.

---

### Note · Authors · 2025-11-12

I have read and agree with the venue's withdrawal policy on behalf of myself and my co-authors.